# Accelerated drug development using a digital formulator and a self-driving tableting data factory

Faisal Abbas [1,8], Mohammad Salehian [1,8], Peter Hou [1], Jonathan Moores [1], Jonathan Goldie[1], Alexandros Tsioutsios [1], Theo Tait [1], Victor Portela [2], Quentin Boulay[3], Roland Thiolliere[3], Ashley Stark[4], Jean-Jacques Schwartz [4], Jerome Guerin[4], Andrew G. P. Maloney [5], Alexandru A. Moldovan [5], Gavin K. Reynolds [6], Jérôme Mantanus[7], Catriona Clark[1], Paul Chapman [2], Alastair Florence [1] & Daniel Markl [1] ✉

Advances in drug discovery and clinical research have shifted the bottleneck in medicines development to chemistry, manufacturing, and controls activities, a critically step for regulatory approval. This includes formulation and process development of a new drug product, which traditionally requires extensive resources, often leading to suboptimal outcomes. These development processes must adapt to follow the advances in drug discovery and clinical research and ultimately shorten timelines while ensuring product quality and safety. In this work, we present an integrated platform for tablet formulation and process development that couples a digital formulator, an in-silico optimisation tool using a predictive material-to-tablet model, with a self-driving tableting data factory, which applies Bayesian optimisation within an automated, fully integrated per-tablet manufacturing to testing workflow. The results demonstrate a reduction in the time from material characterisation to in-specification tablets to 6 h and a reduction in API material use by 65% compared to current state-of-the-art methods.

A wave of artificial intelligence (AI) enabled technologies[1,2] for drug discovery and clinical trials are transforming the way new medicines are discovered and evaluated for efficacy and safety. AI-native drug discovery companies generated an average annual drug pipeline growth rate of around 36% from 2010 to 2021[3], showing an exponential rise of candidates coming to clinical trials. Most global pharmaceutical companies partner with AI technology providers to accelerate drug discovery and clinical research to exploit emerging digital technologies in drug target identification[4,5], generative molecular design[6], automating discovery[7], clinical study protocol optimisation[8], selection of optimal subpopulations[9], dose optimisation[10], therapeutic drug monitoring and dynamic personalised therapy[11], and reducing adverse drug reaction[12]. It is estimated that these scientific advances can shorten drug development timelines from ≈12–15 years to 3-4 years[13]. Advances in drug discovery and clinical research have shifted the primary bottleneck in efficiently delivering new medicines to patients. In particular, chemistry, manufacturing, and controls (CMC) activities, crucial for regulatory approval, now represent a key step on the critical path to registering new medicines. CMC includes the development of manufacturing routes and processes, formulation, scale-up and a

¹CMAC, Strathclyde Institute of Pharmacy and Biomedical Science (SIPBS), University of Strathclyde, Glasgow, UK. ²Glasgow School of Art, Glasgow, UK. ³Medelpharm, ZAC des Malettes, Beynost, France. ⁴DEC Group, Chemin du Dévent 3, Ecublens, Switzerland. ⁵The Cambridge Crystallographic Data Centre, Cambridge, UK. ⁶Sustainable Innovation & Transformational Excellence (xSITE), Pharmaceutical Technology & Development, Operations, AstraZeneca UK Limited, Macclesfield, UK. ⁷UCB S.A, 60 Allée de la Recherche, Brussels, Belgium. ⁸These authors contributed equally: Faisal Abbas, Mohammad Salehian. ✉e-mail: daniel.markl@strath.ac.uk

robust control strategy that reliably provides quality products to patients. This development process involves numerous dependent decisions to transform a new drug candidate into a final product that can be manufactured at scale and meets the target product profile (TPP). From a drug product for first in human (FIH) clinical trials to commercialisation, the financial and time penalties of changing a decision rise exponentially over the course of development using current procedures[14]. CMC development processes must therefore adapt to follow the advances in drug discovery and clinical research and ultimately shorten timelines while ensuring product quality and safety. Digitalisation of CMC processes by the utilisation of digital tools such as big data generation, predictive modelling and AI, automation and robotics can help accelerate drug development timelines by reducing the experimental burden and increasing lab efficiency.

Roughly two-thirds of medicines are administered orally, and approximately half of these medications are in the form of a tablet[15]. Small molecule drugs continue to represent the majority of the pipeline with 52% drugs (29 out of 55) approved by the U.S. food and drug administration (FDA) in 2023 classified as small molecule[16], in line with the five-year average for this modality[17]. Most small molecule active pharmaceutical ingredients (APIs) coming through the pipeline lack desirable raw material characteristics such as good flowability, compressibility, compactability and solubility that are necessary for ease of manufacture and meeting the TPP targets[18,19]. This puts additional pressure on making informed decisions on formulation, manufacturing route and processes.

Process and formulation design follow a cycle of hypothesis generation, experimental design, lab testing, and data interpretation to understand the effects of various formulation and process factors and design a consistent, high-quality product. Yet, this process is often inefficient, repetitive, and time-consuming, potentially spanning months to years. Recent advancements in predictive modelling have been applied across CMC activities, with a particular focus on direct compression (DC) of pharmaceutical tablets, due to its simplicity, cost-effectiveness, and suitability for heat- and moisture-sensitive APIs[20]. DC eliminates the need for granulation steps, thereby reducing production time and preserving the integrity of sensitive compounds[21]. However, formulators often consider granulation due to DC's high dependency on the material properties of the API and excipients (e.g., flowability, compressibility, and compactability). This challenge can be addressed through an autonomous, resource-efficient approach using industrial digital technologies aligned with the manufacturing classification system (MCS)[18,22,23] to proactively scope the likelihood of using DC as a viable processing option for a given formulation to meet the TPP.

Mechanistic models - including first-principle and empirical[24-27] - have been utilised to predict tablet formulation and processing parameters. While these models provide valuable insights into the underlying physical and chemical processes, they require extensive experimental data and in-depth physics-based (mechanistic) knowledge of the problem to estimate their parameters accurately. This demands significant effort in preparing and calibrating formulations and tablets with often partial domain knowledge. Data-driven models, leveraging machine learning[28-31], deep learning, and computer vision[32-35], offer alternative approaches by identifying complex patterns within experimental data. However, the data processing and training/testing pipelines often lack generalisability. This highlights the need for a hybrid approach that integrates mechanistic understanding with data-driven techniques, minimising experimental burdens while predicting the drug product properties directly from raw material attributes[36].

Another critical aspect in the development process is the systematic optimisation of decision parameters, including formulation compositions and process configurations, to achieve the desired TPPs. This requires researchers to navigate the complex interplay of variables, making informed decisions that balance quality attributes, regulatory requirements, and manufacturing efficiencies[37,38]. Bayesian optimisation methods powered by Gaussian processes have been tested in pharmaceutical process engineering due to their computational efficiency[39,40]. However, they are typically used for process optimisation cases with a limited number of decision parameters and may not effectively handle larger-scale problems with numerous variables and different types of input features[41,42]. Therefore, advancing modelling decision-making strategies by combining current methodologies with more robust approaches, such as gradient-based[43,44] and gradient-free[45,46] optimisation methods, is crucial for addressing the correlation between different types of decision variables in tablet formulation and manufacturing processes.

Self-driving labs have gained significant attention in recent years for their potential to revolutionise material discovery[47,48]. These labs leverage robotics, automation, and AI to accelerate discovery and development processes while reducing human error. The closed-loop workflow in a self-driving lab is generally a dynamic operation that cycles through a design, make, test and analyse (DMTA) workflow[49-53]. Relevant examples include a fully autonomous solid-state powder X-ray diffraction workflow using multipurpose collaborative robots[54], self-driving synthesis[55], discovery of heterogeneous catalysis materials[56], and an AI-Chemist platform capable of autonomously performing chemical research tasks, including literature review, experiment design, execution across 14 workstations, and data analysis using machine learning and Bayesian optimisation[57].

Despite this rise in self-driving labs and predictive systems for molecular and material discovery, the development of formulations and process conditions for oral drug products remain a largely manual process requiring a significant time investment of subject matter experts[47,58-60]. While recent advancements in self-driving labs have predominantly focused on liquid-phase systems, particularly in the context of flow chemistry[61] and automated optimisation of organic reactions[62], the development of autonomous platforms capable of handling solid materials remains comparatively underexplored. Existing literature has demonstrated advanced capabilities in optimising reaction conditions, reagent selection, and process parameters in continuous liquid flow systems. However, translating these self-driving principles to systems involving powders and other solid matter introduces additional complexities such as precise dosing, transport, compaction, and real-time characterisation of particulate systems.

In this study, we address a key gap in drug product development by integrating a digital formulator with a self-driving tableting data factory, aligned with the principles of quality by digital design (QbDD)[63], reducing the development time to 6 h and a reduction in API material use by 65% compared to the current state-of-the-art method. The platform workflow begins with defining the specifications for the new drug product for a given API (input information in Fig. 1), including a target drug loading and manufacturability criteria. It then proceeds through two integrated components: the digital formulator and the tableting data factory. The digital formulator employs a hybrid system of models and in-silico optimisation to select the optimal combination of excipients (filler/binder 1 and 2), their respective mass fractions, and initial compaction pressure. The objective is to maximise the flowability of the blend while satisfying a key manufacturability constraint: achieving a minimum tablet tensile strength at specified porosity. Although these constraints can be adjusted, they were set in this study to a tensile strength > 2 MPa at 15% porosity. The tableting data factory is a physically and digitally integrated cyber-physical system for automated dosing, transport and characterisation of powder, and tablet manufacturing and testing, i.e., the make & test system. The orchestration system coordinates the physical and digital tasks, including the data acquisition, processing and management across all stages of the automated make & test system. Once the optimised formulation is established by the digital formulator, the

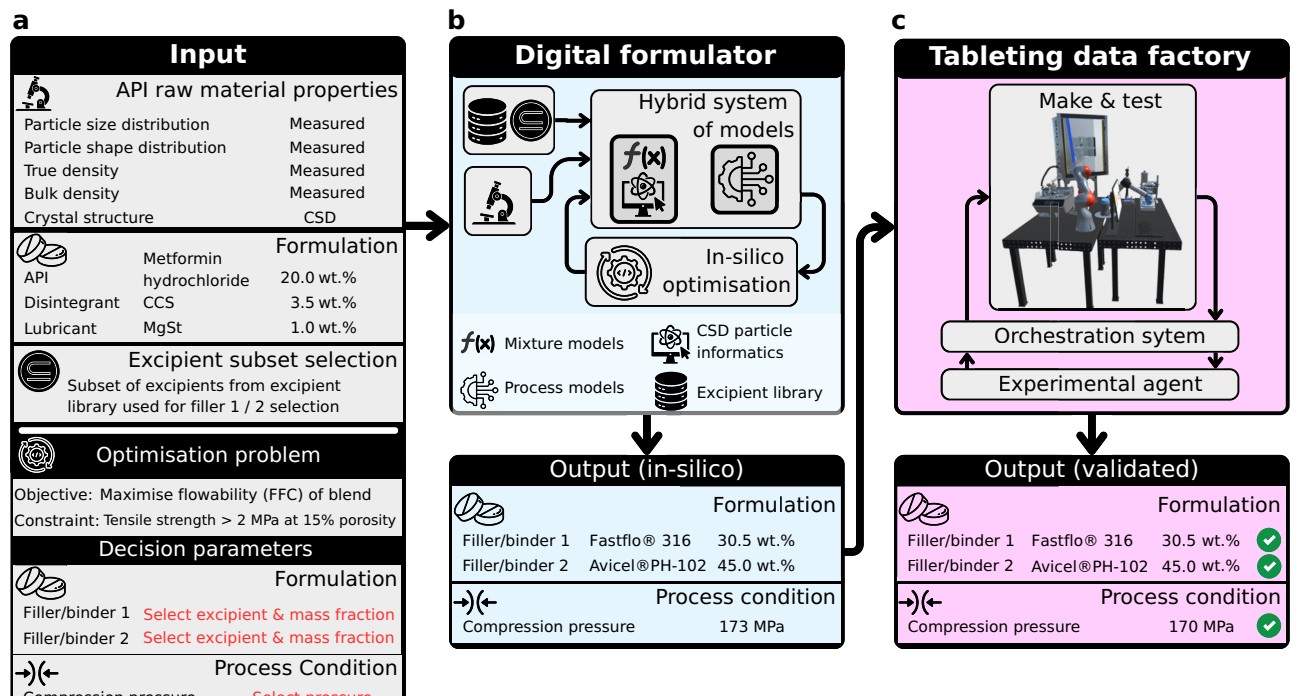

**Fig. 1 | Platform workflow illustrating the input information including the optimisation problem that feeds into the digital formulator and subsequently into the tableting data factory. a** The API raw material properties must be available, together with the drug loading, the disintegrant and the lubricant alongside their respective mass fractions in the formulation. The input also has an option to select a subset of available excipients, based on API-excipient compatibility assessment (e.g., through digital assessment[70]). **b** The outputs of the digital formulator, especially the optimised formulation (filler/binder selection and mass fractions), serve as the direct input to the tableting data factory. **c** The final compaction pressure identified through the experimental agent (Bayesian optimisation) is then validated to ensure the final product satisfies the defined manufacturability constraint. The workflow is exemplified using metformin hydrochloride as a model API.

tableting data factory applies an experimental agent, a Bayesian optimisation approach, to refine the compaction pressure and validate the selected formulation against the defined manufacturability constraint.

Each component of the platform is independently validated using separate experimental protocols and materials. The fully integrated workflow (Fig. 1) is then demonstrated across nine uses cases, involving six distinct APIs including one API evaluated at four different drug loadings.

## Results
### Digital formulator: hybrid system of models
The system of models (Fig. 2) is comprised of two connected steps: (1) mixture models[36] and particle informatics[64], and (2) process models. The mixture models predict the true density, bulk density, tapped density, particle size distribution, aspect ratio distribution and flow function coefficient (FFC) of a blend of materials for a given formulation (API and excipients with their mass fractions) from the raw material's true density, bulk density, particle size distribution and aspect ratio distribution. The material properties of excipients are available in the excipient library with 32 grades of excipient materials that include various grades of microcrystalline cellulose (MCC), mannitol, lactose, dicalcium phosphate anhydrous (DCPA), croscarmellose sodium (CCS), and magnesium stearate (MgSt). A subset of these materials was used for the training and validation of the hybrid system of models. The process models utilise the output of the mixture models with additional input features about the API (i.e., the particle informatics descriptors and API concentration) and the compaction pressure. The mixture models were trained and validated in Salehian et al.[36], while this study provides additional validation data ($R^2 = 0.93$ and RMSE = 0.013), comprising 220 data points across 44 formulations, for the FFC model (Supplementary Fig. 6).

Two ensemble sets of deep neural networks (DNNs) were trained to predict the porosity and tensile strength of tablets from blend material properties, API concentration, and compaction pressure. The blend material properties for both training and validation were predicted using the mixture models with the raw material attributes as input features. Both DNNs were trained and validated on a dataset comprised of 1199 tableting data points from 170 different formulations across 15 different APIs and nine different binders/fillers. Model training was conducted using data from 113 formulations, including a placebo and four different APIs, resulting in 653 data points collected under varying compression pressures. Model validation was performed on an independent dataset consisting of 57 formulations across 11 different APIs, comprising 546 data points obtained under varied compression pressures. Placebo formulations were used exclusively for model training, and no API grade included in the training set was used in the validation set. Both DNN models showed strong performance, achieving $R^2$ values of 0.90 for porosity and 0.89 for tensile strength. The ensemble learning strategy used to train the DNNs allows the prediction of the uncertainty, i.e., the standard deviation of predicted values over the ensemble of models. While higher standard deviations of porosity predictions are associated with higher porosity values (i.e., regions with sparse training data[65]), the trend for uncertainty predictions of the tensile strength is less consistent (Fig. 3). This discrepancy can be attributed to the inherently more complex and nonlinear relationship between input features and tensile strength[66,67]. Tensile strength is more susceptible to both aleatory uncertainty (e.g., inherent variability in material behaviour) and epistemic uncertainty (e.g., limited knowledge), which can compound mixed behaviour in the estimation of uncertainty even when using robust ensemble methods[68].

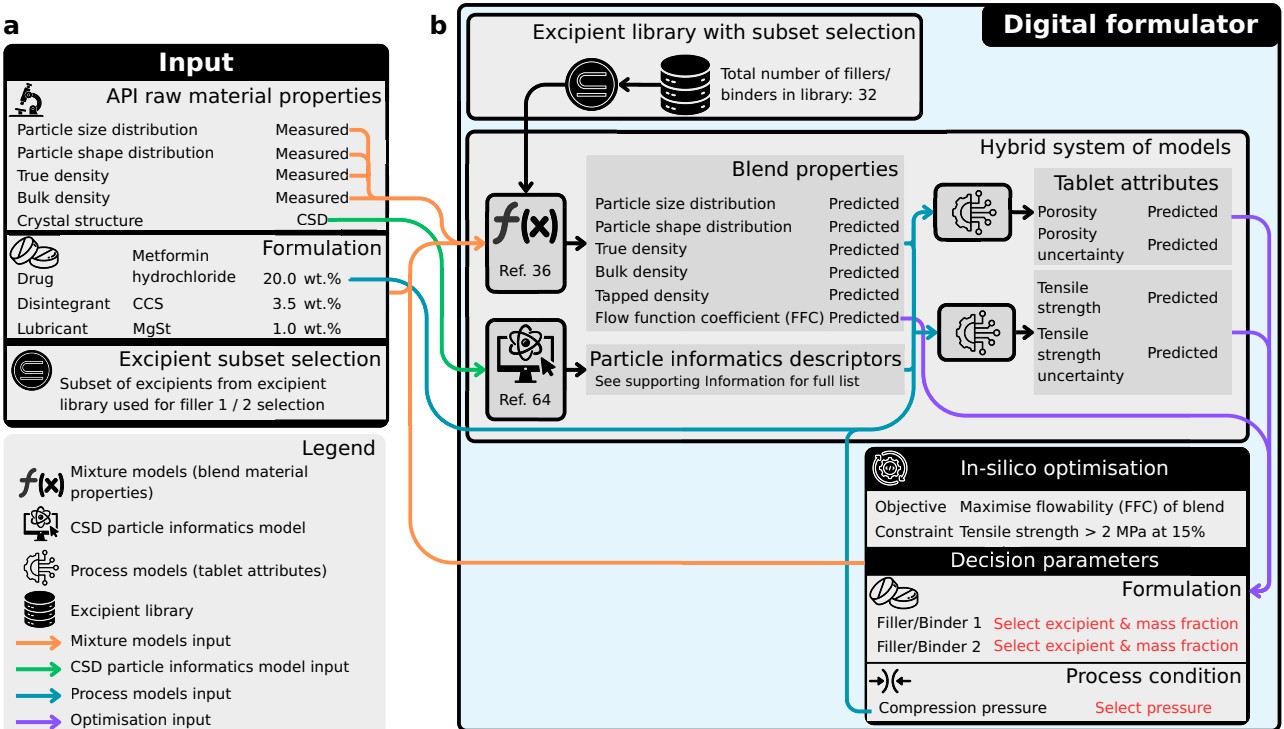

**Fig. 2 | Schematic overview of the digital formulator comprised of the hybrid system of models and in-silico optimisation framework. a** The input consists of the measured API raw material properties, the formulation that includes the drug, disintegrant and lubricant together with their given mass fractions, and the excipient subset selection that is used for the optimisation in the digital formulator. **b** The digital formulator includes the hybrid system of models, the excipient library and the optimisation routine. The excipient library contains the material properties data for all excipients. The excipient subset selection acts as a filter, allowing the user to restrict the in-silico optimisation to excipients chosen based on prior knowledge. However, this set is optional as the algorithm can also evaluate the full set of excipients if no subset is specified. The figure is exemplified using metformin hydrochloride as a model API.

Critically, the modelling framework is designed to be material-agnostic, using exclusively fundamental material characteristics (bulk density, true density, particle size and shape distributions) rather than material identity, enabling generalisation to new materials provided their properties fall within the property space used to train the models (Supplementary Figs. 1–5).

## Digital formulator: in-silico optimisation

The digital formulator uses the hybrid system of models in an optimisation framework (Fig. 2) to find the optimal set of filler/binder 1 and 2, their respective mass fractions, and the initial main compression pressure for a given API and target drug loading, and given disintegrant and lubricant material choices and mass fractions. The objective is to maximise the FFC subject to a manufacturability constraint, i.e., satisfy a minimum tablet tensile strength at a given porosity.

The modular architecture of the digital formulator allows subject matter experts to define exclusion criteria for excipients (set by the user in the excipient sub-selection) based on prior knowledge. For example, incompatible API-excipient and excipient-excipient combinations excluded using their simplified molecular input line entry system (SMILES)[69] representations with compatibility models[70,71] or through experimental screening results[72,73]. This expert-guided approach ensures that only compatible material combinations proceed through the optimisation workflow, reducing the likelihood of selecting a formulation that may exhibit physical or chemical instabilities[74,75].

## Tableting data factory

Following the output from the digital formulator, the platform workflow advances to the tableting data factory (Fig. 1). The tableting data factory comprises of three key components: make & test, an orchestration system and an experimental agent.

## Tableting data factory: make & test

This setup integrates commercially available devices with several customised features (Fig. 4b). Robotic arm 1 (R1) is employed for material transport and to physically interconnect multiple devices. Each iteration of the tablet production begins with the autonomous acquisition of a near-infrared (NIR) spectroscopic dark scan and a reference scan, using a 99% reflectance disc permanently attached to the R1 gripper. The orchestration system captures the reference scan once R1 positions the disc on the NIR spectrometer. Subsequently, R1 proceeds to the dosing unit to acquire the dose for a single tablet. The dosing unit, preloaded with a premixed powder blend, dispenses the precise amount of powder required for one tablet into a customised 3D-printed transportation unit (TU) that is placed on a balance (Supplementary Note 3.1.2). The first decision point (D1) concerns the weight of the obtained powder, measured in real-time (Fig. 4a). Any dose deviating by more than ±5% from the target is rejected and subsequently recycled. The powder dose, that satisfies the weight thresholds moves to the NIR spectrometer stage, where the spectrum is acquired through a sapphire window which is installed at the bottom of the TU. The acquired spectrum is used to monitor the blend homogeneity through monitoring Hotelling's $T^2$ analysis (Supplementary Note 4.3.1). R1 then transports the TU to the compaction simulator, depositing the powder into a 9 mm round-shaped die. Tablets formed in the compaction simulator are conveyed to an automated tablet tester via a customised chute. The automated tablet tester conducts destructive testing on selected tablets, measuring weight, thickness, diameter, and breaking force to determine tablet porosity and tensile strength. The number of tablets undergoing

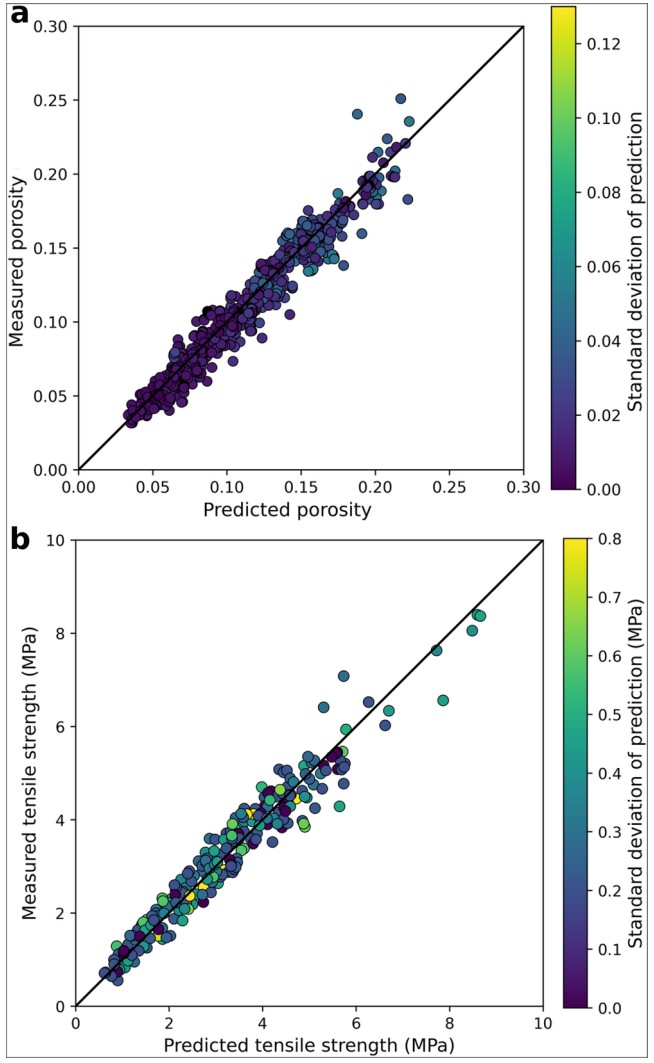

**Fig. 3 | Prediction performance of the hybrid system of models. a** Porosity ($R^2 = 0.90$, RMSE = 0.01) and **b** tensile strength ($R^2 = 0.89$, RMSE = 0.40) predictions for the DNN models using the validation data. The scatter plots compare the measured versus predicted values, while the colour bars represent the standard deviation of predictions. Source data are provided as a Source Data file.

destructive testing is set in the experimental protocol by the user. The remaining tablets undergo non-destructive testing, which measures all parameters except breaking force. A customised tablet separator (TS) then segregates the damaged from the undamaged tablets (Supplementary Note 3.1.4).

The second decision point (D2) assesses whether tablets should be discarded if they fail to meet quality standards such as tablet weight, porosity and tensile strength beyond acceptable variations (± 5%). Undamaged tablets are collected by the robotic arm 2 (R2) using customised gripper fingers and stored in designated containers. Finally, to prevent cross-contamination with other blends, a customised 3D-printed cleaning unit (CU) is used to thoroughly clean the tube and casing of the TU (Supplementary Note 3.1.5).

Benchmarking and validation of the system is focused on assessing 1) the powder and tablet weight, and any related powder loss caused by the powder handling and transportation, and 2) the consistency of repeated experiments.

The primary objective in the design of the TU was to discharge the powder precisely while keeping the powder loss in a consistent range across different formulations. Therefore, powder loss between the powder obtained, measured by the balance, and the final tablet weight

was assessed across ten different formulations and three different target tablet weights (200, 300, and 400 mg) (Supplementary Note 3.1.8). The total powder loss during transportation, including losses from powder dosing and spillage when opening the TU gate in the compaction simulator, consistently stays within 10–22 mg across different formulations and dose weights (Supplementary Note 3.1.8). Material is primarily lost in bulk rather than selectively. Therefore, the ratio of API to excipients remained stable throughout the process. Moreover, the relative standard deviation in powder obtained and the tablet weight remains below 3%. The mean deviation from target tablet weight and the porosity for the different formulations tested was below 4% and 1%, respectively, demonstrating an acceptable consistency across these parameters.

## Tableting data factory: orchestration
The orchestration system serves as the central intelligence of the entire setup, enabling seamless integration of all instruments, coordinates the interaction between various hardware components, and ensures that each instrument performs its function according to a predefined experimental workflow (Supplementary Note 3.1.7). In addition, the orchestration system is responsible for real-time data acquisition and logging, allowing for synchronised data flow across the process chain. The orchestration system not only integrates and controls all instruments but also serves as the platform where the entire experimental workflow is designed and managed. This unified environment allows users to define the sequence of operations, set parameters for each stage, and establish decision-making logic within a single interface.

## Tableting data factory: experimental agent
The primary experimental agent used in this study was a physics-informed Bayesian optimisation (PIBO) framework. PIBO optimises the main compression pressure to meet the target porosity while taking the existing physics-based (PB) correlations between the input and objectives into account, leading to faster convergence and reduced number of experiments required to generate the compressibility and compactability profiles. The PIBO framework aims to optimise the main compression pressure to achieve the target porosity and tensile strength while satisfying the underlying (empirical) compressibility and compactability models that provide prior information about the relationship between main compression pressure, porosity, and tensile strength. The experimental agent is fully customisable and can be adapted for a range of applications; an additional example illustrating its use for scale-up assessment is provided in Supplementary Note 3.2.3.

## Demonstration of platform workflow
The platform workflow (Fig. 1) was applied to nine case studies (Fig. 5), comprising five different APIs while one API was studied across four different drug loadings. For each case study, the raw material properties of the API were measured. Across all case studies, the lubricant and disintegrant material and their mass fractions were kept constant with CCS at 3.5 wt.% and MgSt at 1 wt.%. We selected a subset of the excipients (Fig. 5a) for the optimisation, which align with common materials to ensure findings are relevant to real-world applications[76,77]. It should be noted that the hybrid system is inherently flexible and can readily accommodate an expanded list of excipients as appropriate. The optimisation problem was set up to maximise the FFC of the blend at a consolidation pressure of 1.6 kPa with the constraint of meeting > 2 MPa of tensile strength at a 15% porosity tablet. These constraints are based on literature recommendations[18,22] to deliver a tablet that has a sufficient strength while it is also porous enough to ensure fast liquid uptake and, consequently, effective disintegration of the tablet as required for immediate release formulations.

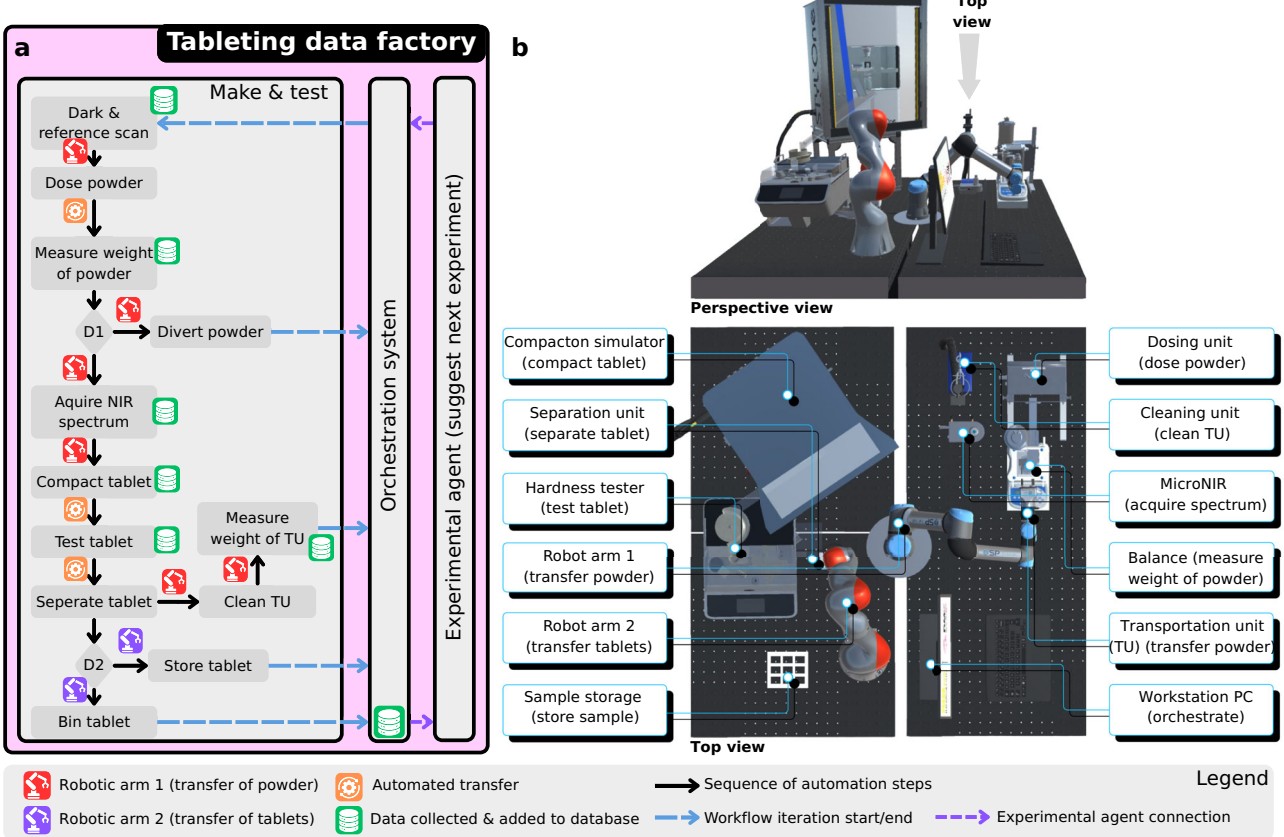

**Fig. 4 | Overview of the tableting data factory. a** The make & test workflow detailing the operation of the automated system alongside the automated data management and experimental agent. D1 is the first decision point to either discard the powder dose or take it forward based on the weight of the dose. D2 is the second decision point to either discard the damaged or unacceptable tablets or move them to storage. **b** Perspective and top view of the setup indicating the location of each instrument on the table. Supplementary Movie 1 demonstrates the operation of the tableting data factory.

For each case study, the digital formulator was run to optimise the selection and mass fraction of filler/binder 1 and 2 as well as the initial main compression pressure (Fig. 5b). In all cases (except GR 20 wt.% where Lactose FastFlo 316 has the higher concentration), single or multiple grades of MCC dominate the optimal solutions, demonstrating its significant role in achieving the required tensile strength within the imposed constraints which is attributed to its superior compactability properties[77,78]. Conversely, other excipients such as Lactose Granulac 200 M and Mannitol Pearlitol 200 SD are either minimally utilised or entirely absent from the optimal formulations, indicating their limited contribution to maximising the FFC under the given manufacturability constraint. The sporadic, concentration-dependent inclusion of Lactose FastFlo 316 and MCC Avicel PH101 suggests that their presence in the formulation is highly sensitive to the specific API and its loading. There are, however, multiple other factors that needs to be further considered when optimising the formulation. For example, the proportion of MCC can be potentially restricted due its insoluble nature, which can affect the tablet's dissolution rate and API bioavailability[79]. It also has a high moisture content, posing stability issues for moisture-sensitive APIs[80]. Moreover, MCC is strain rate sensitive, which can lead to inconsistencies in tablet strength during high-speed manufacturing[81]. Therefore, despite its advantages, it's important to balance MCC's use by considering these factors and potentially exploring alternative excipients to address these challenges. Another parameter is the impact of lubricant and disintegrant on the manufacturability and performance (e.g., disintegration, dissolution) criteria, which remains a promising topic to be studied in future work.

The comparison between predicted and measured FFC values (Supplementary Fig. 25) shows that predicted FFC values generally trend higher than measured values, particularly for Aspirin (AS; 20 wt.%) and Griseofulvin (GR; 20 wt.%), attributable to the model error and the uncertainty of flowability data, especially in higher FFC values. Despite being lower, the measured FFC values of all optimal formulations were validated through the preparation and characterisation of the respective blends with no flowability issues observed during the processing confirmed through the assessment of powder loss and weight uniformity (Supplementary Fig. 39a). In addition, the validation of the FFC model with an expanded dataset reveals consistent performance to its original version (Supplementary Fig. 6). This suggests that, while the predictive model may pose prediction errors, it effectively guides the formulation process towards producing robust and viable tablets. While there are inherent errors in predictions, the model's prediction accuracy remains superior compared to the state-of-the-art using both regression[82] and classification[83] approaches.

The PIBO framework was tested with the tableting data factory across the nine formulations identified by the digital formulator to achieve a target porosity of 15%. During the optimisation process, the tuning parameters of the physics-based models varied initially, then converged to a plateau after several iterations (Fig. 5c). The termination criteria (based on the change in the goodness of fit of empirical compressibility and compactability models to the collected data) were activated during different iterations based on the speed of convergence in each case study. The termination of optimisation is followed by a validation experiment at the target porosity, where a compression pressure is suggested based on the physics-based models with the identified tuning parameters.

The measured porosity is compared with the predicted target to evaluate the accuracy of calibrated models. The digital formulator, on

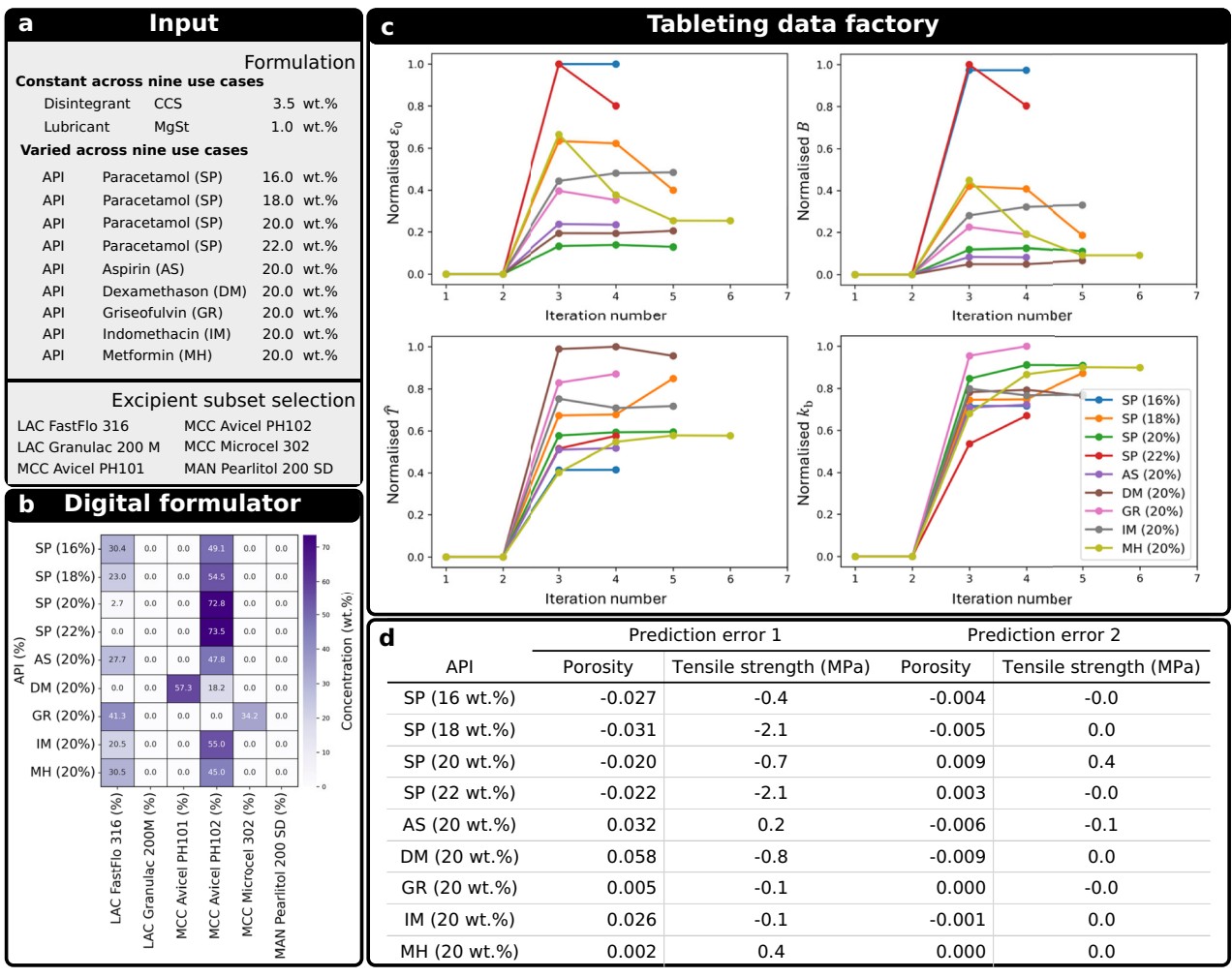

**Fig. 5 | Demonstration of the platform workflow across nine use cases using six different APIs. a** Summary of formulation case studies and excipient subset selections. **b** Summary of results from formulation optimisation cases. The *x*-axis represents different case studies with target APIs and their corresponding mass fractions (wt.%). The *y*-axis lists the subset of excipients considered in the formulation optimisation process. The colour scale indicates the mass fraction (wt.%) of each excipient. The '0.0' indicates that this excipient was not chosen by the digital formulator for the given use case. **c** Variation of normalised values of tuning parameters of Kawakita (a: $\varepsilon_0$, b: *B*) and (c: $\hat{T}$, d: $k_b$) Ryshkewitch-Duckworth models during the PIBO cases for the different optimised formulations. Each tuning parameter is normalised to the range of 0 and 1 using the minimum and maximum values observed across all iterations. **d** Summary of the prediction errors. Prediction error 1: The absolute error between the initial model-based prediction (using digital formulator) and the experimental result at the optimal compression pressure. Prediction error 2: The absolute error between the post-calibrated Gaussian process regression model in the PIBO and the experimental result at the optimal compression pressure (validation point). Negative values indicate underestimation, while positive values represent overestimation. Colour intensity indicates deviation from zero. Source data are provided as a Source Data file.

average, overestimated the porosity by 0.26% and underestimated the tensile strength by 0.63 MPa (Fig. 5d). The comparison between the initial prediction by the digital formulator and calibrated models after PIBO demonstrates that the initial predictions (Supplementary Figs. 37, 38), are relatively close to the final calibrated models. While the PIBO calibration process further refines the predictions, the adjustments required for the initially predicted profiles by the digital formulator require only 4 − 6 experiments for the case studies investigated in this work, underscoring the reliability of the DNN-based process models in making a close-to-optimum first-time prediction of tablet attributes from raw material properties. The general underestimation of the initial predictions is due to the conservative definition of the objective function to justify the optimisation in scenarios where extensive experimentation may not be feasible, and a right-first-time formulation optimisation is required.

The assessment of blend homogeneity was performed qualitatively using Hotelling's $T^2$ analysis (Supplementary Fig. 36), which evaluates the multivariate distance of each spectrum from the principal component analysis (PCA) model centre to identify any deviations between iterations. This approach confirms the blend sub-samples do not deviate significantly from one another, supporting the assumption that the blend is homogeneous. Across all case studies, the tablets met the manufacturability constraint ( > 2 MPa at 15% porosity) and satisfied the disintegration requirement in less than 15 min (Supplementary Fig. 42), in line with expectations for immediate release tablet formulations.

## API particle size impact on manufacturability
The hybrid system of models in the digital formulator can be used to generate a new understanding of the effect of various physical properties, such as particle size distribution (PSD), on blend flowability and tablet tensile strength. To demonstrate this, we systematically examined different types of PSDs (e.g., unimodal and bimodal) with distinct characteristics to elucidate the relationship between PSD and blend flowability (FFC) and tablet tensile strength. Understanding these relationships is crucial for particle engineering and formulation

scientists seeking process insights to optimise API particle characteristics and tablet properties.

We investigated 16 synthetic APIs with distinct particle size distributions: 8 unimodal (UMD) and 8 bimodal (BMD) distributions. The distributions were synthetically generated by parametrising log-normal distributions (Supplementary Note 4.2.2), covering a range of values across the parameter space used to train and validate the hybrid system of models. Generated distributions were used as the API PSD to simulate blend and tablet properties using the hybrid system of models. All other properties of the API (e.g., true and bulk densities, aspect ratio distribution, particle informatics descriptors) were kept identical to paracetamol (SP). The formulation used for simulation consisted of API at varying drug loadings, 3.5% of CCS, 1% of MgSt, and equal concentrations (1:1 ratio) of Lactose FastFlo 316 (LAC1) and MCC Avicel PH102 (MCC2), followed by a 100% drug loading as pure API. Simulations were run for each PSD at different drug loadings, varying from 0 (i.e., placebo) to 100% (i.e., pure API) with 20% increments in drug loading. For each case, flowability of the formulation (FFC at 1.6 kPa) as well as the porosity and tensile strength within a range of compression pressure from 100 MPa to 400 MPa with 50 MPa increments were predicted, followed by interpolating the tensile strength at 15% porosity through fitting the Ryshkewitch-Duckworth model[84] to the simulation data. Supplementary Tables 11 and 12 list the user-defined parameters to generate UMD and BMD, denoting the synthetically generated unimodal and bimodal distributions, respectively. The increasing integer in UMD1 to UMD8 and BMD1 to BMD8 reflects a progressive increasing in mean particle size and/or span. Supplementary Tables 13 and 14, respectively, summarise the simulation outputs for flowability and tensile strength.

The systematic investigation of particle size distribution effects on FFC (Fig. 6a) shows that finer particle distributions (e.g., UMD1-UMD2) exhibit significant reductions in FFC as drug loading increases, decreasing from 6.5 to 1.2 and 6.6 to 1.4, respectively, crossing into the very cohesive flow regime (FFC < 2)[85,86]. The exceptional flow improvement noted for UMD7, BMD7, and BMD8 can be attributed to two key factors: (1) reduction in specific surface area, minimising cohesive interactions[87], (2) improved packing efficiency through enhanced void structures that improve powder mobility[88,89]. Accordingly, UMD7's exceptional improvement in FFC can be attributed to a lower percentage of fine particles (as detailed in Supplementary Table 11, showing the percentage of particles below a specified size threshold), despite UMD8 possessing a larger median particle size ($D_{50}$). These mechanisms are also supported by studies of granular flow in pharmaceutical powder systems, where the presence of larger particles reduces interparticle friction and facilitates improved flow[90,91].

Figure 6b shows the results for tensile strength at 15% porosity using unimodal and bimodal API PSDs at different drug loadings. For all particle size distributions, tensile strength showed a pronounced decline as drug loading increased. This trend aligns with the established understanding that higher API concentrations typically reduce tablet strength due to the poorer compactability of crystalline APIs[92,93]. A marked decline occurs when drug loadings exceed 20%, providing model-based confirmation of the established empirical understanding that percolation thresholds are typically low for many API materials in direct compression processes[18]. This computational validation of the ≈ 20% threshold aligns with industry experience that higher drug loadings create formulation challenges due to the formation of percolating networks that compromise tablet mechanical properties[94,95].

At drug loadings up to 60%, bimodal API PSDs consistently outperformed unimodal distributions, yielding slightly higher tensile strength at 15% porosity values, an advantage that could be attributed to the improved packing efficiency and interparticle bonding facilitated by the presence of both fine and coarse API particles[96,97]. However, at very high drug loadings (80% − 100%), this advantage

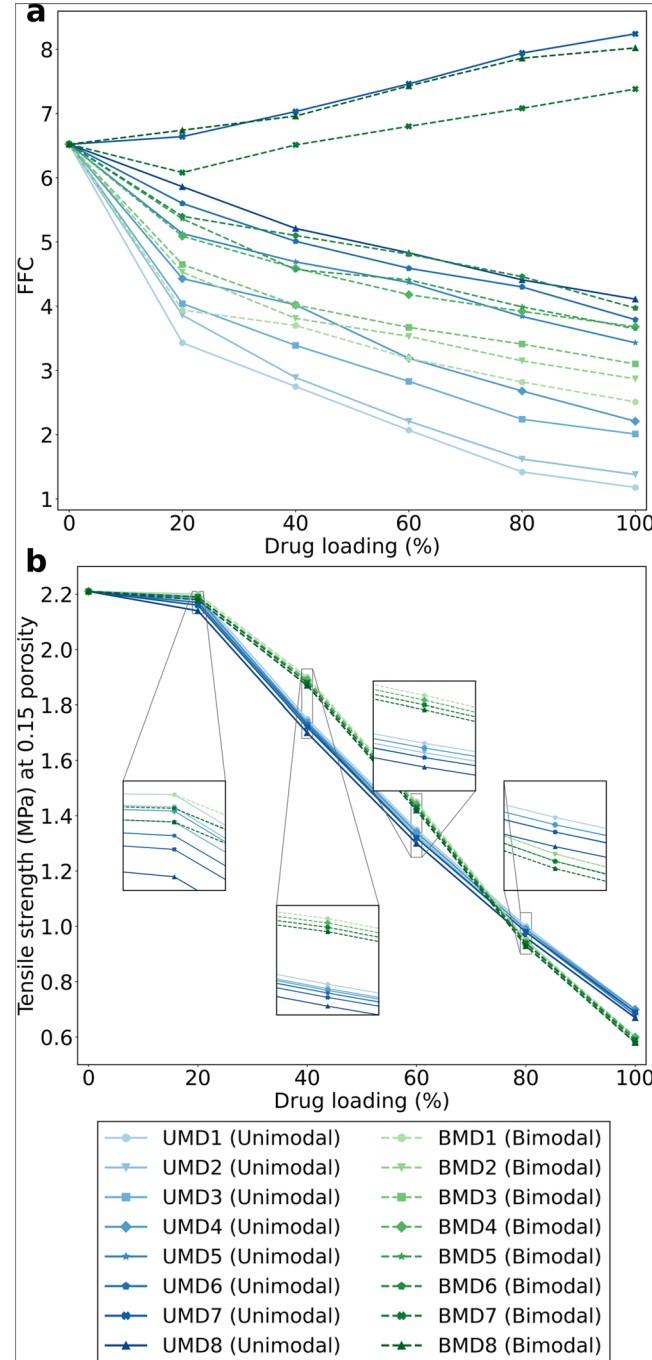

**Fig. 6 | Predicted manufacturability characteristics for varying API particle sizes. a** FFC at consolidation pressure of 1.6 kPa and **b** tensile strength at 15% porosity for 16 synthetic API particle size distributions at different drug loadings. UMD and BMD denote the synthetically generated unimodal and bimodal distributions, respectively. The increasing integers in UMD1 to UMD8 and BMD1 to BMD 8 reflect a progressive increasing in mean particle size and/or span. Source data are provided as a Source Data file.

diminished and eventually reversed, with unimodal API PSDs surpassing bimodal systems in tensile strength.

The impact of distribution type is significantly smaller than that of the drug loading on the tensile strength. However, some trends can still be observed across both distribution types. Smaller particle sizes generally resulted in higher tensile strength at 15% porosity at all drug loading levels. For unimodal distributions, formulations with $D_{50}$ of 10 μm (e.g., UMD1, UMD2 in Supplementary Table 11) maintained higher

tensile strength than those with $D_{50}$ of 150 µm (e.g., UMD8 in Supplementary Table 11). This effect can be attributed to the increased surface area available for interparticle bonding with smaller particles, leading to stronger compacts[98]. Similarly, narrower particle size distributions (i.e., lower span values) generally resulted in tablets with higher tensile strength compared to wider distributions at the same $D_{50}$. For example, UMD1 showed higher tensile strength than UMD2 at all drug loading levels. This can be attributed to more efficient particle packing and greater interparticle contact area in narrower distributions[99,100].

## Sensitivity analysis of optimised formulations

A sensitivity analysis was conducted on the excipient ratios of the optimised formulations generated by the digital formulator. For each blend, the model ensemble predicted tensile strength at a fixed porosity of 15% and the FFC (Supplementary Figs. 26–34).

Across lactose-containing formulations, higher lactose fractions consistently reduced tensile strength at 15% porosity while increasing FFC, showing the intrinsic trade-off between the optimisation objective (maximising blend flowability) and the constraints on tensile strength and porosity for mechanical integrity[29]. The analysis also provides quantitative justification for the selected optima. For instance, the optimal formulation in the SP 16% case satisfies the tensile-strength criterion up to ≈ 30 wt.% lactose; beyond this threshold, tensile strength declines and its predictive uncertainty rises sharply, whereas FFC reaches a plateau at ≈ 45 wt.% lactose. Similar trends are observed for SP 18%, SP 20%, and SP 22%, where FFC plateaus at high lactose levels, but tensile strength falls below the acceptance limit once FFC increases abruptly. In AS 20% and IM 20%, the optimiser could propose higher lactose loadings to further improve FFC; however, the corresponding drop in tensile strength suggests that these solutions would violate the tensile strength constraint. Model-based uncertainty behaves as expected: predictive variance for FFC grows with increasing FFC values, consistent with the behaviour reported for the initial model version[36]. Uncertainty in tensile strength predictions is lower at moderate strengths but becomes heterogeneous at the extremes, reflecting the complex, nonlinear coupling among excipient properties.

## Feature importance analysis

Mean SHAP values[101] were calculated for both porosity and tensile strength models to enhance interpretability and understand the relative importance of input features (Supplementary Figs. 13 and 14). Compression pressure, representing process settings, exhibited the greatest impact on both predicted outputs, while inclusion of particle informatics descriptors showed minimal impact on models' prediction. Tablet porosity demonstrated the second-highest influence on tensile strength prediction, confirming alignment with established empirical compactability models[84]. Tablet porosity demonstrated the second-highest influence on tensile strength prediction, confirming alignment with established empirical compactability models[84]. True density and PSD principal component 1 (PC1), representing inherent mixture properties, showed considerable impact on both outcomes, highlighting the importance of assessing these parameters' effects on tablet manufacturability and quality through computational modelling.

## Tableting data factory for manufacturing

The tableting data factory can be operated in a manufacturing mode, delivering small-scale batch products for a specific set of specifications. While the current setup is not designed for good manufacturing practice (GMP) compliance, it serves as a proof-of-concept for applications such as early-phase clinical trials, dose titration studies, or personalised healthcare, where flexibility and adaptability are critical[102–104]. Real-time process monitoring is facilitated by augmented and mixed reality (AR/MR) visualisation tools (Supplementary Movies 2 and 3 and more details in the Supplementary Note 5),

enabling parameter monitoring and adjustment to ensure high-quality output in real-time scenarios[105]. Real-time process monitoring is facilitated by augmented and mixed reality (AR/MR) visualisation tools, enabling parameter monitoring and adjustment to ensure high-quality output in real-time scenarios[105]. Real-time process monitoring is facilitated by augmented and mixed reality (AR/MR) visualisation tools, enabling parameter monitoring and adjustment to ensure high-quality output in real-time scenarios[105].

## Benchmarking and performance metrics

It is important to note that: 1) no existing model in the literature predicts tablet properties directly from raw material attributes without requiring prior blend or tablet preparation for calibration, and 2) no fully automated experimental workflow currently exists at this scale. As a result, our comparison is limited to specific cases where a meaningful benchmark could be established.

Specific performance metrics of the platform, including the digital formulator and the tableting data factory are ref. [106] with a comparison to state of the art where applicable:

Throughput and productivity: The data factory achieved a tablet production rate of 72 tablets per hour, encompassing powder quality monitoring, manufacturing, testing, storage, and data handling. A continuous run over 24 h would deliver ≈ 1440 tablets, accounting for periodic interruptions such as cleaning cycles, hopper refilling, or brief maintenance, thus averaging 60 tablets per hour. State-of-the-art: in drug product development, compaction simulators are most commonly used with either manual feeding (≈ 10 tablets per hour) or a shoe / paddle feeder (≈ 20 tablets per hour)[107,108].

Material usage: A comparison with state of the art approaches demonstrates a 65% reduction in API material usage with our platform compared to the approaches reported in the literature (Supplementary Note 6). The platform workflow requires less than 2.65 g of API for the raw material characterisation and the tableting data factory using the PIBO framework to make tablets with a weight of 250 mg. The absolute amount of API material usage depends on several factors, including the tablet weight, number of experimental repeats, and drug loading. It should be noted that the majority of material usage in this platform arises from raw material characterisation, which is performed only once and can subsequently be used to predict blend and tablet attributes for any formulation State-of-the-art: The material usage of the best state-of-the-art approach, as detailed in the Supplementary Note 6, is ≈ 7.5 g[109]. This approach requires calibration and, hence, would need to be repeated for blends with a different excipient.

Human resource requirements: The automated workflow requires a total of 6 h distributed across material characterisation (4 h), formulation optimisation via the digital formulator (0.5 h), blend preparation (1 h), and tableting using the data factory with PIBO (0.5 h). State-of-the-art: Human resource requirements are greater, with a total of around 14 h needed for material and API characterisation[26,107], manual data handling, predictive tool use[110], blend preparation, and compaction pressure optimisation. These time estimates do not include the blend characterisation, e.g., flowability. It should be noted that these estimates are based on industry experience due to the lack of reported data in the literature.

Degree of autonomy: The automated system operated autonomously during the formulation optimisation and tableting process, requiring human input only for blend preparation and hopper filling. State-of-the-art: At present, no autonomous platform exists that addresses this specific application.

Operational lifetime: The system maintained continuous operation with refill of the dosing hopper required after every 100 tablets to avoid powder accumulation and ensure dosing accuracy. In addition, cleaning of the compaction punch was necessary after ≈ 1000 tablets. State-of-the-art: Existing systems do not support continuous operation across all stages.

Accessible parameter space: The data factory explored a broad design space, including dose weight, pre- and main compaction pressures, dwell time, compaction profiles, and multiple methods for NIR and tablet testing. State-of-the-art: Existing systems can access the same parameter space using manual material and data handling.

## Discussion

This study introduces a new approach to tablet formulation and process development, offering a significantly more resource and time-efficient alternative to current state-of-the-art methods. Compared to conventional approaches, the system achieves API material savings of ≈ 65% and reduces development time by around 60% for a given blend. Benchmarking of the tableting data factory demonstrated high consistency in powder and tablet weight, minimal powder loss during automated handling, and good repeatability under fixed process conditions. By minimising manual intervention, the approach enhances accuracy, precision, and consistency across all stages of production, establishing a robust and scalable platform for pharmaceutical tablet development.

Beyond material and time efficiencies, this platform contributes to generating new understanding across material, blend and tablet attributes. The combination of the tableting data factory producing high-quality, structured data that feeds into the digital formulator enables systematic knowledge generation and supports informed decision-making in formulation development. This was demonstrated through the use of the predictive models, to investigate the effect of unimodal and bimodal PSDs on flowability and tablet tensile strength across drug loadings. The simulations confirm known but previously anecdotal trends, including reduced flow with finer PSDs and improved flow with bimodal size distributions. A sharp decline in tensile strength beyond ≈ 20% API provides model-based validation of the percolation threshold observed in practice.

While the digital formulator demonstrably accelerates formulation and process development, several constraints merit acknowledgement that provide opportunities for methodological advancement and expanded applicability. The porosity and tensile strength predictive models are trained on a finite library of APIs and excipients, creating potential extrapolation risks when encountering novel chemical entities or processing conditions beyond the encoded parameter space – an existing challenge with the majority of data-driven predictive models. This work addressed this limitation through strategic diversification across varied chemical classes and systematic validation against test APIs, whilst future expansions could encompass more materials with finer particle sizes, and alternative tablet sizes and geometries beyond the current 9 mm format. Further steps could include the design of a model maintenance framework that triggers the augmentation of the model with new materials and data. The ensemble approach, while providing uncertainty quantification through standard deviation estimates across predicted porosity and tensile strength values, assumes homoscedastic noise characteristics and may inadequately capture heterogeneity arising from batch-to-batch variability or scale-dependent manufacturing perturbations. Future implementations can incorporate heteroscedastic noise modelling approaches to provide more principled uncertainty quantification by directly modelling aleatoric uncertainty. The uncertainty in training data may also affect SHAP importance rankings due to the absence of standard deviation measurements. This limitation could be addressed by incorporating input parameter distributions into the training process, such as including variance and expected values in the loss function. Furthermore, the framework inherits limitations from the underlying FFC mixture model's dependence on shear-cell measurements, which exhibit sensitivity to environmental factors such as ambient moisture and consolidation history, potentially propagating errors throughout the validation and optimisation workflow; whilst comprehensive material characterisation achieved strong predictive performance ($R^2 = 0.93$), future developments should integrate multiple flowability assessment techniques beyond shear-cell testing and real-time monitoring of environmental conditions during material handling. In addition, the particle informatics descriptors, derived from static crystallographic databases, may not adequately represent mechanical characteristics affecting the powder processing; although the nine selected descriptors enhanced tensile strength prediction from $R^2 = 0.86$ to 0.89, integration of process-structure-property relationships through multi-scale modelling approaches represents promising pathways for capturing real-time morphological changes during processing.

The tableting data factory, while highly automated and efficient, presents certain operational limitations. It is currently restricted to a fixed round-shaped die with a minimum of 9 mm diameter for powder compaction, limiting its flexibility in accommodating different tablet geometries and smaller sizes. To maintain the dosing accuracy, the system's maximum blend holding capacity is 100 grams, necessitating manual top-up after each 100-gram batch, which may interrupt continuous operation for extended runs. Routine maintenance is also required, particularly cleaning of the compaction punch after every 1000 tablets to prevent buildup and ensure consistent tablet quality. In addition, while the system maintains high precision in tablet weight and tensile strength, some variability is observed in porosity measurements, likely due to limitations in sensing resolution.

A major challenge in CMC development is the scale-up of processes from laboratory-scale to commercial scales. This transition often introduces scale-dependent issues, such as over-lubrication, segregation, sticking, lamination/capping and flow problems. The current platform addresses some of these challenges by maximising flowability of the blend to minimise the risk of flow-related problems and by assessing key scale-up parameters, pre-compression pressure, main compression pressure and dwell time (Supplementary Note 3.2.3), to mitigate lamination and capping risks. While the current version does not explicitly assess risks of over-lubrication, segregation, or sticking, the modular design of both the digital formulator and the tableting data factory enables future integration of models to capture these additional scale-up effects.

## Methods

### Materials

This study used 26 different materials, including 18 different API grades and six fillers/binders, one disintegrant and one lubricant. All materials were used as received. Details regarding the material IDs, characteristics of the excipients and APIs can be found in Supplementary Tables 1 and 2.

### Digital formulator: hybrid system of models

PCA was performed to reduce the dimensionality of the predicted particle size and aspect ratio distributions by mixture models into two sets of three principal components[111]. The API descriptors include its concentration to explicitly model the effect of drug loading on tablet attributes, and the crystallographic and particle informatics descriptors represent properties relevant to the processing and mechanical behaviour of the API (Supplementary Table 4)[112]. Table 1 summarises the input features used to train process models. For the tensile strength model, the response variable was log-transformed (i.e., the natural logarithm of tensile strength, $\ln(\sigma_t)$) to incorporate knowledge on the exponential relationship between compression pressure and tensile strength[107].

To evaluate the models' ability to generalise to new APIs, a leave-API-out approach was employed in splitting the dataset into training and test sets. Specifically, the validation data was composed by 57 different formulations (amounting to 546 data points) containing Paracetamol (SP), Griseofulvin (GR), Aspirin (AS), L-Ascorbic Acid

**Table 1 | Input features for the process models**

| ID | Parameter | Size | Descriptor | Source |
|---|---|---|---|---|
| 1 | Mixture true density | 1 | Blend property | Mixture model |
| 2 | Mixture bulk density | 1 | Blend property | Mixture model |
| 3 | Principal components (PCs) of particle size distribution* | 3 | Blend property | Mixture model |
| 4 | PCs of aspect ratio distribution* | 3 | Blend property | Mixture model |
| 5 | Tapped density | 1 | Blend property | Mixture model |
| 6 | Flowability (FFC) | 1 | Blend property | Mixture model |
| 7 | Main compression pressure | 1 | Process condition | Process settings |
| 8 | API concentration (drug loading) | 1 | Formulation | Formulation |
| 9 | Particle informatics descriptors** | 9 | Calculated particle properties | CSD Python API[65] |

*PCA was applied to the predictions of particle size and aspect ratio distribution mixture models, and the first three principal components was used as inputs to the process models.
**The list of particle informatics descriptors is provided in Supplementary Table 4.

(ASC), Ciprofloxacin Hydrochloride (CIPH), Doxycycline Hyclate (DOX), Guaifenesin (GF), Indomethacin (IM), Levetiracetam (LVCT), Metformin Hydrochloride (MH), Dexamethasone (DEX), three Ibuprofen grades (IBU2, IBU3, IBU4). The training data contained 113 formulations (amounting to 653 data points), including placebo tablets and tablets with Benzoic Acid (BZ), Lovastatin (LOV), one ibuprofen grade (IBU1), and Mefenamic Acid (MF). It is worth noting that while IBU1 was included in the training dataset, three additional ibuprofen grades (IBU2-4) with distinct physical properties were retained in the validation dataset to assess the model's generalisability, aligning with the material-agnostic nature of the framework, which relies on fundamental material properties rather than material identity. This strategy was designed to assess the models' performance in predicting tablet attributes for APIs not seen during training. Supplementary Table 3 summarises the tablet data based on the API, their usage in training or validation, and the range of drug loading, porosity, tensile strength, and compression pressure.

Following the ensemble learning methodology outlined in Salehian et al.[65], an ensemble of DNNs (with 20 models per ensemble in this study) was trained in different random seeds, with the final output for porosity or tensile strength being the average of the individual models' outputs. This ensemble modelling strategy enhances the robustness of the DNN by mitigating the effects of random initialisation on training performance and enabling the estimation of the standard deviation for future predictions – an important measure for assessing the model's prediction quality for new data points. All DNNs were trained using the same dataset and model architecture, which consisted of two hidden layers with 128 units, each followed by a ReLU activation function.

To mitigate the risk of overfitting, several safeguards were implemented: (1) the ensemble learning approach reduces overfitting by averaging out model-specific biases and provides uncertainty estimates through the standard deviation of predictions[113,114]. (2) A leave-API-out validation strategy was used to evaluate generalisation performance to previously unseen APIs, which provides a more stringent test of model robustness than conventional random splitting. (3) The DNN architecture was kept simple with only two hidden layers of 128 units each, following ReLU activation functions, to limit model complexity relative to the available data and balances model expressiveness with the risk of overfitting[115,116]. 4) Early stopping callbacks were utilised to prevent overfitting and optimise training duration (Supplementary Figs. 11, 12)[117]. The convergence of training and validation losses for both DNN-based porosity and tensile strength models confirmed successful regularisation without overfitting[118]. In addition, the superior performance of DNNs over Random Forest and Support Vector Regression in the leave-API-out validation (Supplementary Table 5) suggests that the ensemble DNN approach successfully captures generalisable patterns rather than memorising training-specific features.

## Digital formulator: in-silico optimisation

The in-silico optimisation utilising the hybrid system of models is defined as:

$$
\begin{aligned}
J_{\substack{x \in \mathbb{R}^{N_x} \\ m \in \mathbb{R}^{N_m}}}(x, m) &= -\text{FFC} \\
&\text{Subject to :} \\
\theta_{\hat{\sigma}} - \left[E(\hat{\sigma}) - \alpha \times \delta_{\hat{\sigma}}\right] &< 0 \\
\theta_{\hat{\epsilon}} - \left[E(\hat{\epsilon}) - \beta \times \delta_{\hat{\epsilon}}\right] &< 0
\end{aligned}
\tag{1}
$$

where $x$ is the $N_x$ dimensional vector of decision variables; $m$ is the $N_x$ dimensional state vector of raw component properties (e.g., particle size and aspect ratio distribution, true density, bulk density); $E(\hat{\epsilon})$ and $E(\hat{\sigma})$ are the expected mean value of the predicted porosity and tensile strength, respectively; $\theta$ is the user-defined threshold for manufacturability conditions ($\theta_{\hat{\sigma}} = 2$ MPa and $\theta_{\hat{\epsilon}} = 0.15$ in this study), $\delta_{\hat{\epsilon}}$ and $\delta_{\hat{\sigma}}$ are the standard deviation of the predicted porosity and tensile strength, respectively; $\alpha$ and $\beta$ are user-defined constants (both are set to 0.2 in this study) to define the allowable level of risk in the robust optimisation process. Higher values of these risk factors result in a more conservative optimisation, thereby reducing the likelihood that the formulation will fail validation. Non-dominated sorting genetic algorithm II (NSGA-II)[119] was used as the optimisation algorithm due to its proven capability in global search and independence from calculating the gradient. The population size and the number of iterations were set to 30 and 50, respectively.

## Tableting data factory: make & test

The tableting data factory setup was built on an M6 tapped table spanning $200 \times 200$ cm$^2$. R1 and R2 have a reach of 850 and 820 mm and can carry up to 5 and 14 kg of load, respectively. The FlexPTS (DEC Group, Switzerland) technology is used to dose the pre-mixed powder blend. The quantity of the powder discharged from the dosing unit is volumetric based where the volume can be adjusted by setting the height of a piston altering the powder chamber height. The dosing unit collects the powder in the chamber using a vacuum, the powder is discharged using compressed air. The controller of the dosing unit has the capability to change the duration of the vacuum pump and the pressure of the compressed air to dispense powders with different physical properties, e.g., to consider variations in density, and particle size/shape.

The R1 gripper releases the TU on the weighing balance (Cole-Parmer PA-224I, United States) that is placed under the dosing unit. The tube that carries the powder in TU has the capacity of 3000 mm$^3$ (Supplementary Note 3.1.2). A sliding gate operated through a linear solenoid is used to hold and then release the powder dosed from the dosing unit. To perform NIR measurements for blend homogeneity

assessment, the gate of the TU incorporates a 1 mm thick, 10 mm diameter sapphire glass window. The round tube that holds the powder within TU has a diameter of 8 mm. As this small tube can pose a challenge when discharging adhesive and cohesive materials, the inside of the tube was coated with PTFE to create a non-stick surface and a motorised vibrator, which is activated upon the opening of the gate. Two customised 3D printed fingers are mounted on the robotic gripper that grasp the TU from the back to transport the powder to the different stations (Supplementary Note 3.1.3). As the mass of the powder discharged into the TU is weighed prior to tablet manufacturing, the electronic devices in the TU need to be electrically connected with fingers through metallic touchpoints to allow the TU to remain connection-free and standalone when it is placed on the weighing balance. Through this connection, the electric solenoid and the vibrator are operated by an external and customised electronic control unit. The control unit for the TU receives the control commands from the orchestration system via serial communication and operates the solenoid and vibrator based on an external power supply. The orchestration system also acquires the initial weight of the TU before getting the dose and subtracts this weight from the final weight to determine the true value of powder obtained in that iteration.

A NIR spectrometer (Micro NIR PAT-W, VIAVI, United States) is incorporated in the workflow to assess blend homogeneity. As changing ambient conditions may influence the NIR measurement, new dark and reference scans need to be acquired over time. Therefore, a 99% reflectance disc is attached to R1 to take the reference scan at the beginning of each iteration.

Tablets are produced using a compaction simulator (STYL'One Nano, MEDELPHARM, France). The door of the compaction simulator was replaced by a laser curtain to provide the robotic arm R1 easy access to the compaction die for powder discharge. The compaction simulator's control system halts the entire machine immediately if the laser curtain is interrupted. It then waits until R1 has dispensed the powder and moved clear of the laser curtain before initiating the compaction process. The tablet chute is kept within the boundaries of the laser curtain and linked with the tablet tester through a side wall to avoid any process interruption.

A fully automated tablet tester (AT50, SOTAX, Switzerland) is used to measure breaking force, weight, diameter, and thickness of the tablets. The testing process starts as a tablet enters the feeder of the tester, where it moves through various stations to assess its properties. A bespoke TS is used to differentiate between damaged and undamaged tablets. The flow of tablets is controlled by adjusting the position of a motor-driven barrier (Supplementary Note 3.1.4). To ensure proper alignment of the undamaged tablets, a linear solenoid pushes them onto their flat surface. Bespoke robotic fingers (Supplementary Note 3.1.3) are mounted on the R2's robotic gripper to accurately grip the tablet for transport.

The CU is attached to a high-power vacuum cleaner that is operated by an external control unit. The operation of CU is signalled by the orchestration system, and it only runs when the TU requires cleaning.

### Tableting data factory: orchestration system

The orchestration system, designed in LabVIEW, can control and monitor all instruments remotely (Supplementary Note 3.1.7). Each instrument is assigned a dedicated computer for two main reasons: 1) to provide local access for users who may need to operate the instrument separately for other tasks, and 2) to standardise diverse communication protocols into a unified protocol for seamless communication with the orchestration system (as illustrated in Supplementary Note 3.1.7). The orchestration system also communicates with the experimental agent.

### Tableting data factory: experimental agent

The compressibility model developed by Kawakita[120,121] and the compactability model developed by Ryshkewitch-Duckworth[84,122] were used in PIBO as the empirical models due to their proven capability in capturing the compression profile[27]:

$$\varepsilon(P) = \frac{\varepsilon_0}{1 + \left(\frac{V_\infty}{V_0}\right)bP} = \frac{\varepsilon_0}{1 + BP} \tag{2}$$

$$\sigma(\varepsilon) = \hat{T}e^{-k_b\varepsilon} \tag{3}$$

where $\varepsilon$ is the porosity of the tablet, $\sigma$ is the tensile strength of the tablet, and $P$ is the main compression pressure. For the Kawakita model, $\varepsilon_0$ is the initial powder bed porosity, $V_\infty$ is the net volume of powder, $V_0$ is the initial apparent volume of powder, and $b$ is a tuning parameter which is hypothesised to reflect the resistant and cohesive forces of the particles[123]. Following Tait et al.[27], the ratio of volumes $\frac{V_\infty}{V_0}$ and the constant $b$ were grouped into the single tuning parameter $B$ to simplify the fitting process. For the Ryshkewitch-Duckworth model, $\hat{T}$ is the strength at zero porosity, and $k_b$ represents the material's bonding capacity. The acquisition function in the classic (black-box) Bayesian optimisation is constrained such that:

$$\text{Err}(f_K(\underline{X}), \mathcal{D}_n) \leq \text{Err}(f_K(\underline{X}), \mathcal{D}_{n-1}) \tag{4}$$

$$\text{Err}(f_{R-D}(\underline{X}), \mathcal{D}_n) \leq \text{Err}(f_{R-D}(\underline{X}), \mathcal{D}_{n-1}) \tag{5}$$

where $f_K$ and $f_{R-D}$ are the Kawakita and Ryshkewitch-Duckworth models, respectively; $\mathcal{D}_n$ refers to the dataset of collected experimental observations at iteration $n$; $\text{Err}(f(\underline{X}), \mathcal{D}_n)$ denotes the error (root mean squared error in this study) between the model and the collected data at iteration $n$. These constraints mean that while minimising the acquisition function from black-box BO, the optimisation trajectory is confined to reduce the error between the collected data and models. The optimisation is terminated if the change in the goodness of fit within the last two iterations falls below a user-defined minimum threshold (in this study, set to be less than 20% change in the tuning parameters fit, i.e., $\text{Err}(f_K(\underline{X}), \mathcal{D}_n)$ and $\text{Err}(f_{R-D}(\underline{X}), \mathcal{D}_n)$, as compared to the previous iteration, i.e., $\text{Err}(f_K(\underline{X}), \mathcal{D}_{n-1})$ and $\text{Err}(f_{R-D}(\underline{X}), \mathcal{D}_{n-1})$). These criteria were established in the acquisition function to ensure the tuning parameters of compressibility and compactability profiles are calculated with sufficient accuracy, enabling reliable prediction of the profile across a continuous range of input parameters.

### Reporting summary

Further information on research design is available in the Nature Portfolio Reporting Summary linked to this article.

## Data availability

The data that support the findings of this study are available from the Source Repository[124] and are from the corresponding authors upon request. Source data are provided in this paper.

## Code availability

Codes are available from the Source Repository[124] and from the corresponding authors upon request.

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

## Acknowledgements

The authors thank the Digital Medicines Manufacturing (DM²) Research Centre Co-funded by the Made Smarter Innovation challenge at UK Research and Innovation (Grant Ref: EP/V062077/1) for funding this work. The authors also thank the EPSRC ARTICULAR project (Grant ref: EP/R032858/1) and EPSRC Future Continuous Manufacturing and Advanced Crystallisation Research Hub (Grant Ref: EP/P006965/1) for the data generated and exploited in this work. This work was also supported by Research England and the Scottish Funding Council under the UK Research Partnership Investment Fund Net Zero Medicines Manufacturing Research Pilot.

## Author contributions

F.A., M.S., A.F., and D.M. contributed to the conceptualisation of the project. M.S. developed the digital formulator with input from D.M. F.A. developed the automation of the tabletting data factory with

contributions from A.T., P.H., and D.M. F.A. developed the orchestration software for the tableting data factory. J. Moores, J. Goldie and T.T. contributed to the data generation for the digital formulator. F.A., J. Goldie and A.T. contributed to the data generation on the tableting data factory. Q.B. and R.T. contributed to the modification and digital integration of the compaction simulator. A.S., J.J.S., and J. Guerin contributed the digital integration of the dosing unit. A.G.P.M., A.A.M., and M.S. contributed to the calculation of the particle informatics descriptors and their integration into the digital formulator. G.K.R. and J. Mantanus contributed to defining the objectives and constraints for the workflow demonstrations. V.P. developed the XR application with input from P.C., C.C., and F.A. F.A., M.S., and D.M. drafted the initial manuscript, and they revised the manuscript with all authors' input. D.M. supervised the project.

## Competing interests

G.K.R. reports a relationship with AstraZeneca that includes: employment and equity or stocks. Apart from this, the authors declare no competing interests.
