## [Transparent Peer Review file · Nature Communications]

Accelerated drug development using a digital formulator and a self-driving tableting datafactory

Corresponding Author: Professor Daniel Markl

Version 0:

Reviewer comments:

Reviewer #1

(Remarks to the Author)

The application of AI, digitalization, and automation holds immense potential to accelerate the development of new drugs. Research in this vital area can significantly impact the health of patients worldwide. This paper represents an important first step in developing a self-driving tableting station. The authors have demonstrated novel contributions in three main areas:

1. Digital Formulation of Blends: This approach ensures the production of tablets with optimal quality attributes.
2. Self-Driving Tableting Station: This system can orchestrate multiple tasks to convert blends into tablets efficiently.
3. AI-Driven Control and Optimization: The use of AI model to control and optimize the tableting process.

These innovations make the manuscript worthy of publication in Nature Communications. However, before I can recommend its publication, I would like the authors to address a few key points listed below:

1. Compatibility of Excipient and AI: The Digital Formulator is missing a crucial element—the compatibility of excipients with the new AI. This is the first step in designing a formulation and should be addressed before focusing on other attributes like blend flowability and tablet tensile strength.
2. Scale-Up Challenges: The main challenge in the CMC development process is scaling up from bench scale to pilot and commercial scales. Powders often exhibit physical effects like over-lubrication, segregation, and flow issues such as jamming and arching, which may not be present at small scales. Typically, early-stage industrial development is carried out with an eye towards larger scale, and appropriate scale models are developed to bridge between scales. The formulation and process optimization approach presented here may be sub-optimal at a larger scale.
3. Tablet Shapes, Thickness, and Tooling: The paper and supplementary material do not mention tablet shapes, thickness, and tooling used to manufacture tablets. Model predictions of process parameters like pre-compression and main compression pressures to achieve target tablet tensile strength and porosity will depend on the tablet shapes and tooling used. This information is crucial for accurate model predictions and should be included.
4. Porosity Constraint: The porosity constraint (> 15%) used for in silico optimization is counterintuitive. It will end up over-constraining the space for feasible solutions. The usual industrial practice is to increase the solid fraction or reduce porosity until the desired tablet hardness or tensile strength is achieved.
5. NIR Method Development:
 - o How much effort went into developing the NIR method for assessing blend homogeneity and tablet content uniformity? This should be included in Table 2.
 - o NIR spectra are known to depend on API particle size and blend bulk density. How was the baseline spectral drift corrected?
 - o Was the NIR model recalibrated to accommodate changes in API particle size and blend bulk density?

Minor Points:

1. On line 238, the mass fractions of CCS and MGST have been flipped.
 2. The correct Y-axis label for Figure S12(b) should be 'Tablet Porosity (%)'.
 3. It is unclear if section 7.4 on 'Extended Reality' in the main paper adds any scientific value. It should be moved to the supplementary information to help reduce the length of the paper.
- Addressing these points will strengthen the manuscript and provide a clearer understanding of the platform's capabilities and limitations.

(Remarks on code availability)

Reviewer #2

(Remarks to the Author)

This study presents an integrated platform for tablet formulation and manufacturing, built around a Digital Formulator and a Self-Driving Tableting DataFactory. This is an interesting study. However, the study may be better suitable for pharmaceutical journals.

Several technical issues are to be solved.

1. The dataset consists of 113 tablet formulations with 653 data points collected under different compression pressures. The authors employ deep neural network (DNN) models to predict the porosity and tensile strength of tablets containing previously unseen active pharmaceutical ingredients (APIs). Given the small size of the dataset and the complexity of the DNN algorithm, is there a risk of overfitting?
2. The study only contains several drugs and excipients (e.g. MCC, lactose, CCS, MgS). Although these excipients are widely used in tablet formulations, there are much more other excipients in tablet formulations.
3. How do the small-scale formulations transition to large-scale manufacturing batches?

(Remarks on code availability)

Reviewer #3

(Remarks to the Author)

Authors report a detailed description of an approach for intensifying powder based product design with an example from pharmaceutical products. This piece of work is elegant engineering, programming and automation example, ultimately orchestrating powder handling flow in a (potentially) self-optimizing manner.

However, the reported work remains very 'developmental' focusing on increasing the number of experiments that can be performed, with a limited focus on scientific questions! Current manuscript is focusing on an approach simply aiming to find a (local) optimum composition without asking the fundamental question 'Why?'. Authors do not really go into using this approach to understand the physics of particulate systems better. What are the underlying reasons for powder flow, compaction of particles, strength of a compact – what is affecting charging of particles (particle surface) -> flow, what is the effect of particle size distribution, why does a compacted powder form a solid compact with a high enough tensile strength? Please discuss this properly and strengthen this part in the manuscript.

The second major comment is the target product profile. What about the most critical property of a compacted pharmaceutical tablet for immediate release, disintegration? Authors do not mention disintegration nor measure any disintegration behavior (or very shortly, mainly pointing to 'future studies should...'). It can be assumed that tablet porosity correlates with disintegration time, but this is quite far fetch with the current materials. Please consider including this into the work properly.

Third major comment: state of the art on self-optimizing labs is dominantly with liquid phase system (organic reaction optimization in flow reactors etc). Authors have a point in their work, and they have a clear novelty, because now they operate with powders, solid matter. This important aspect is not really spelled out for the reader. Please compare and contrast properly this aspect of novelty in the Introduction chapter of the work.

Fourth one: authors refer to physics-based models. This is a strong expression for basically curve-fitting (empirical models of Kawakita and R-D). Here it is suggested that authors tune down the physics, especially considering that they are not really discussing the fundamental physics of powder flow and underarticulate interactions affecting tablet processes.

More minor comments, but still something that should be addressed:

The chemical space of both excipients and active pharmaceutical ingredients is quite limited. Basically few excipients (cellulose, lactose, calcium phosphate and mannitol) and six/eight simple APIs. Also, considering the chemical space: authors have an interface to Cambridge Structural Database (CSD). However, very few descriptors are used for eight materials. Considering that CCDC is advertising on their homepage that they have 1.25M accurate 3D structures, the author's discussion seems very limited.

Considering powder flow and tablet compaction, it is well-known that particulate properties are of critical importance. As a continuation of the previous comment on 'chemical space', the 'particulate space' is quite limited: APIs are relatively large particle size and no really challenging APIs are included in the experimental data set.

Please comment on this and revise accordingly.

Particle characteristics are described in Table S1. Particle size is reported with two digits (=10 nm), which is not realistic considering sampling error and the method used for this analysis. And at the end of the day, this is not even relevant for making powders flow and to compact them.

Strengthen the description of particulate space, and discussion related to this.

Same Table S1: textbook is teaching us that MCC Avicel PH101 is closer to 50 microns and PH102 is 100 microns. Here the values are opposite. This is most likely a typo, BUT raising a concern of the overall quality of the raw data. It is appreciated that data is available in the Source Repository, but readers should be able to trust the data in the times when everything is easy to make automated and just copy-paste the errors to the next generation of scientists.

Authors make it difficult to capture the composition of their model system. Line 239 is stating that there is 3.5 w% of MgSt (a really high amount), but Supporting info is saying 1 w%.

Supporting info Line 28: formulated as presented in Table S1?? Please proofread your text.

Throughout the text; authors refer to Figures in the main text and supporting information randomly without always indicating S for Figure in the supporting information. Please proofread the text carefully for that aspect.

Table 2 in Conclusions seems to appear in a wrong place. It can be included in one of the earlier chapter as a discussion element, but it seems wrongly placed when in Conclusions.

Minor minor comment: the use of abbreviation XR might confuse a reader with pharmaceutical background. This abbreviation is often use for extended release (XR) products. Here authors report work on immediate release. Otherwise, very elegant chapter and novel approach within the pharmaceutical community.

(Remarks on code availability)

Reviewing the full code is outside my core expertise and would have taken too much time to do full validation of that. Please find someone with more data science background to do that part. I have read this work from a practical, pharmaceutical science, point of view.

Reviewer #4

(Remarks to the Author)

(Remarks on code availability)

Reviewer #5

(Remarks to the Author)

Reviewer report for Nature Communications manuscript #NCOMMS-25-15811

Manuscript title: Accelerated Medicines Development using a Digital Formulator and a Self-Driving Tableting DataFactory
In this study, the Authors discussed an automated tableting platform that leverages machine learning, modelling tools such as physics-informed Bayesian optimization (PIBO), and robotic automation. The authors offered a material-to-product approach to predict and optimise critical quality attributes for different formulations, linking raw material attributes to tablet properties such as flowability, porosity, and tensile strength. The motivation for the work is important, namely to improve the sustainability and efficiency of direct compaction tablet drug product development. Substantial research and development efforts are noted and applauded for. However, the paper does not represent significant enough scientific innovation. The paper employed mostly established, or the author group's previously established methods, and did not report any scientifically significant findings. While the time and material saving claims are noted, they lacked substantiation. The material and time saving by itself is considered as lack of scientific significance by the reviewer. Automation is naturally expected to speed things up as an incremental engineering advancement rather than scientific innovation. In addition, the writing of the manuscript is significantly below the standards of Nature Communications. The discussion is confusing to such an extent that the scientific quality of data is called into question. The reviewer provides the following review feedback with the intention of being as constructive as possible. The reviewer recommends the authors, after major revisions, to re-submit the manuscript to a more process engineering-oriented journal, potentially as 2 manuscripts. The reviewer recommends rejection of the manuscript from further publication consideration by Nature Communications.

Major concerns on Figures and Tables:

Setting aside for a moment the content, the presentation clarity and quality of the majority of the figures and tables do not meet Nature Communications standards.

Figure 1 was intended to clarify the overarching workflow. Instead, the reviewer found it to cause more confusion. Please clearly annotate and explain:

1. The relationship among the blocks.
2. Which blocks represent measurements, and which represent prediction.
3. The relationship between shaded blocks and bullet texts without shade.
4. The meaning of different greyscale shades of the shaded blocks, if any.
5. The blocks under thick arrow – are they supposed to be the input or output to the main block (e.g., "Material Characterisation" – shouldn't particle size be an output rather than an input?). If the arrows do not imply input, output or directionality, it should not be utilized.
6. Figure 1C was referenced (line 308) but was not annotated in the figure.
7. Why did "Tableting DataFactory" show up twice? Does Material Characterization only input to Digital Formulator, but not to Tableting DataFactory?

It is disappointing to the reviewer that the authors started a Nature Communications manuscript Figure with a pile of bullet points, indicating a lack of architectural deliberation, presentation clarity, or both.

Figure 2 is an expansion of "Digital Formulator" from Figure 1.

1. "Hybrid system of models" block is consistent with Figure 1, but "In-Silico Optimisation" is not. Optimization appeared several times in Figure 1, thus it is impossible to correspond and rationalize how these optimization modules in Figure 2 relate to that of Figure 1.
2. Particle size appeared again several times in "Material (Powder) Space", presumably as the input of "Mixture models". How are they related to the "Material Characterisation" module in Figure 1?
3. Similar question to Figure 1, what properties are measured, literature/database retrieved, or predicted?
4. Is "New API characterization" part of the optimization loop, as the current figure suggests?

Figure 3 – Is prediction accuracy only demonstrated on training data? If so, validation predictions on parameter space that were not part of the training should be shown to allow the reviewer to truly assess the model's predictability. If the validation predictions are already in the plot (16 out of 113 as suggested in the method section), they should be clearly differentiated from the data used in the training. In Figure 3b, there is a range on the left of the histogram, >2.7MPa, that does not have any data. The corresponding range in the cross-correlation plot below showed a lot of data. Please explain. Why are the histograms in Figure 3 and Figure S1 different?

Figure 4 – What is the purpose of rows with all zero values? Discuss or remove.

Figure 5 needs major improvements. The robotic arm icons are very small and difficult to distinguish between #1 and #2. Consider bright colors such as Red and Blue that can help. The flow chart in panel (a) is difficult to decipher; the light gray arrows should have darker gray colors. In panel (b) some of the text is going outside the textbox, which is unacceptable. How is Figure 5 related to Figures 1 and 2? Consistent terminology and colour coding of blocks are essential to present a complex, layered workflow. SCU is referenced repeatedly in the text, but not annotated in the Figure. Modules are mentioned without any sequence, nor annotation, making the narration extremely difficult to follow.

Figure 7 - This figure is overall poorly discussed. The caption referred to two tuning parameters as a and b, while there were no references of the four sub-figures a, b, c, and d. The figure showed a lot of data, however, was only briefly explained in line 402, which neither helped clarify the main point of the figure, nor offered explanation of the data. A plateau was referenced but was far from universal. Some clearly did not reach a plateau (Using Figure 7a as an example, red, orange, pink, and green did not reach a plateau). Why? Termination criteria were briefly discussed, but produced confusing results – why is red clearly not plateaued, but terminated, while grey, brown and green are plateaued in iterations #3 and #4 already, but not terminated and ran to iteration #5? Why does the parameters remain zero for both iterations #1 and #2, and for all the parameters? They cannot be simply explained as “underestimation of the initial predictions”. Rather, some systemic initialization issue is called into concern.

Figure 8 is insufficiently discussed.

Figure S2 is considered by the reviewer as one of the primary scientific findings from this manuscript. It is discussed extensively in the main manuscript body by the authors, while the figure is included as Supporting Information. The inconsistency is an ignorance at best. It should be included into the main manuscript. The discrepancy is substantial, and the explanation is insufficient. 2 out of 9 data points failed the FFC requirements, 4 under predicting, 5 over predicting, and some with error over 50%. This should be not claimed as “This consistency”.

Table 2 should not be in the conclusion section. It needs to be completely redesigned for a journal publication. Table should focus on presenting the numbers, while the text should be incorporated into discussions. All numbers need to be substantiated with data from the paper or literature. For example, it is very difficult to believe the state of the art method is “manual filling” or “shoe feeder”. Shouldn't an automated system intended for Nature Communications be compared with a state of art system used in the industry? If it is compared to a laboratory specific manual filling method, then the benchmark is not state of art, and the claims on wider industrial impact need to be revised. Further, to follow the author's claims that the 72 tablets per hour rate included powder quality monitoring, tablet manufacturing, testing, storage and data handling, should the corresponding time using manual filling include those monitoring, testing, and handling elements as well? The table stated 72 tablets per hour, while the introductory section emphasized 1440 tablets in 24 hours, which is 60 tablet per hour. It is again inconsistency and ignorance at best.

Fundamental Modelling Concerns

- Is 113 tablet formulations (653 data points) considered sufficient? The reviewer has doubt on the massive parameter space that can be representatively covered by 653 data points.
- Can the authors comment how the 653 training data points cover the parameter space of the properties of the 9 formulations predicted? This comment links back to the training and validation comment for Figure 3. The reviewer appreciates the rich set of additional data included in “Support Information”. However, the reader should be able to understand the main points supported by the main manuscript. The supplementary document should be considered as optional, not mandatory.
- What are importance scores of the input parameters? It would simplify the computation if only the relevant input parameters are retained in the modelling.
- An ensemble of Deep Neural Network (DNN), Random Forest and Support Vector Machine (SVM) were compared. However, the claim about DNN being the “superior” model is questionable. Yes, the SSE and R2 values in Table S3 favor DNN slightly. However, DNN tends to overfit and is less reliable in the context of the limited training data. How does the limited training set (only 653 data points to train a DNN for 21 features) impact the different models differently. For example, in Figure S1 tensile panel, there are 3 outliers near 3.2 MPa which does not exist for RF.
- More fundamentally as a study design concern, if predictions were made using full set of input parameters including calculated particle informatics descriptors (Figure 3), DNN's “superior performance” needs to be reevaluated against other models using the additional particle informatics descriptors.
- Lines 223 onward lacks definitions on
 - o Ensemble learning
 - o Ensemble of models
 - o Exploration
 - o Exploitation and how it is different from exploration

The reviewer realizes some of this may be due to Nature Communication's requirement on moving method section after conclusion. It is the authors' responsibility to ensure narration coherency.

- Line 232: Is the sub-set of excipients used for the training, but not the rest of excipients? If it is the case, the other excipients are not relevant and should be excluded from the manuscript entirely. If it is not, then the reviewer has further concerns on the authors' choice in what data was included in the training, what was included in validation, and what was included for the optimization.
- Line 235: Does the expanded list of excipients need to be included in the training?
- Line 249: MCC1 did not have FFC data (Table S1). How does that impact MCC1 being chosen as an excipient?

- Line 373: Porosity and tensile strength have been presented as primary prediction parameters. Why is elastic recovery introduced later? Either make it consistently part of the prediction parameters or elaborate on its delayed inclusion.
- Line 463: GP is not defined. Figure S10 incorrectly referenced.

MR/AR section

This section is both poorly written and poorly integrated. The argument of its innovation and demonstration of its value are both weak. While the reviewer appreciates it as an interesting technology demonstration, it does not belong to this manuscript in general and is very far away from Nature Communications standards.

Grammar, language and consistency issues

These issues are throughout the manuscript with examples noted below,

- In the Abstract: "This approach accelerates... tablet within 6 hours..." should be "tablet TO within 6 hours."
- On page 26 footnote: "The list of informatics descriptors IS"
- Figure 5: To reduce clutter, the abbreviations PIBO and SuMoBO should be deleted. Keep capitalization consistent ("Orchestration" versus "transfer tablets.")
- Figure 6: Typo in panel (a) "DataFacotry"
- In Table 2, "hoper" should be hopper.
- There are typos in Figure S1, "SVR" should be SVM.
- "Table 1" and "Table 1 in Supporting Information" should be better distinguished with Table S1. Similarly for other references to Tables and Figures.
- The Abstract mentions less than 5 grams of API required, however it needs a direct comparison with the "state-of-the-art" e.g. >75% savings in materials. This would highlight the impact of the technology in easily understandable terms.
- Line 506: The reviewer found the use of versions confusing and unnecessary. Given the excessive length of the manuscript, the reviewer suggests removal of any reference to "version", and focus on the latest and most up to date results.
- The word "DataFactory" in the title is confusing. It maybe best to drop it; "Self-Driving Tableter" would be enough.
- Line 315 – what happened to the powder if they are neither diverted to waste nor reused? It is implied that it will be used by the process anyway. Clarify or remove.
- Line 316 – How are the tablets not meeting uniformity criteria identified?
- Line 358 – Is there a mode other than self-driving that the authors intended to compare?
- Line 419 – Figures S8 and S9 have nothing to do with the context. Consequently, the comparison cannot be assessed by the reviewer.

(Remarks on code availability)

Code repository and instructions are appreciated, though they do not change the recommendation.

Version 1:

Reviewer comments:

Reviewer #1

(Remarks to the Author)

I have carefully reviewed the revised manuscript titled 'Accelerated Medicines Development using a Digital Formulator and a Self-Driving Tableting DataFactory'. The authors have addressed most of my critical suggestions and have responded thoughtfully to the comments from other reviewers as well. While a few minor points remain, I am generally satisfied with the revisions and the authors' clarifications.

I believe this paper represents a significant contribution to the field of medicines development, introducing innovative approaches that open up many avenues for future research and design. I therefore strongly recommend it for publication in Nature Communications.

(Remarks on code availability)

Reviewer #2

(Remarks to the Author)

The authors well answered the reviewers' questions.

(Remarks on code availability)

Reviewer #3

(Remarks to the Author)

The author team has properly addressed the comments from my previous review.

(Remarks on code availability)

Reviewer #4

(Remarks to the Author)

(Remarks on code availability)

Reviewer #5

(Remarks to the Author)

The authors' extensive effort in revising the manuscript is recognized. The overall quality has been improved. The enhanced data and presentation elevated the manuscript as 1 or 2 publishable papers in a PharmSci or a process engineer journal. The reviewer recommends a rejection of the manuscript to Nature Communications. In addition to a continued disagreement with the authors on the NatComms worthy innovation, the manuscript falls short in scientific rigor and validated impact required. As the authors incorporated the clarity, visualization, and language issues suggested by the reviewer, the general concerns from the reviewer were either insufficiently addressed or deflected. The inclusion of new data and contents, while intended to address some of the previous issues, created new inconsistencies, a large number of new questions (e.g, the majority of the 43 figures in supplemental material were not clearly explained), and concerning patterns. The issues are too many to enumerate, with a few examples listed below.

As a major conclusion, 65% material reduction has been emphasized in the abstract, main innovation summary, and the first discussion point. The supporting data, however, is presented as the last section of the supplemental material, where the number of tablets manufactured from each method are substantially different. The reviewer disagrees with this conclusion, while concerns on the benchmarking objectivity at a principle level.

While Figure 3 was enhanced with more data, questions and concerns on the original data from the reviewer were ignored. Can the reviewer assume that the authors acknowledged that previous Figure 3 was wrong, and replaced it with the new Figure 3? New Figure 3b increased the range of tensile strength, though new Figure 3a decreased the range of porosity. What happened to the porosity data between 0.3-0.4? Were they thrown away because of a poorer fit? By the same token, should the reviewer question potential manipulation of other data?

Shortened abstract focused mainly on the problem statement with inflated significance that the reviewer finds trouble to agree, while failing to summarize key scientific innovation. This reflects the overarching lack of clear, to the point innovation from this paper. Piling up data and material does not necessarily make it more innovative, if not the opposite.

(Remarks on code availability)

Version 2:

Reviewer comments:

Reviewer #2

(Remarks to the Author)

The authors answered the reviewers' questions.

(Remarks on code availability)

Reviewer #4

(Remarks to the Author)

(Remarks on code availability)

Reviewer #5

(Remarks to the Author)

(Remarks on code availability)

Response to Reviewers for “Accelerated Medicines Development using a Digital Formulator and a Self-Driving Tableting DataFactory”

Faisal Abbas^{1^} & Mohammad Salehian^{1^}, Peter Hou¹, Jonathan Moores¹, Jonathan Goldie¹, Alexandros Tsioutsios¹, Theo Tait¹, Victor Portela², Quentin Boulay³, Roland Thiolliere³, Ashley Stark⁴, Jean-Jacques Schwartz⁴, Jerome Guerin⁴, Andrew G. P. Maloney⁵, Alexandru A. Moldovan⁵, Gavin K. Reynolds⁶, Jérôme Mantanus⁷, Catriona Clark¹, Paul Chapman², Alastair Florence¹, Daniel Markl^{1*}

[^] Authors contributed equally.

* Corresponding author; email: daniel.markl@strath.ac.uk

¹ CMAC, Strathclyde Institute of Pharmacy and Biomedical Science (SIPBS), University of Strathclyde, Glasgow, G1 1RD, UK

² Glasgow School of Art, Glasgow, G3 6RQ , UK

³ Medelpharm, ZAC des Malettes, 615 Rue du Chat Botté, 01700 Beynost, France

⁴ DEC Group, Chemin du Dévent 3, 1024 Ecublens, Switzerland

⁵ The Cambridge Crystallographic Data Centre, 12 Union Road, Cambridge, CB2 1EZ, UK

⁶ Sustainable Innovation & Transformational Excellence (xSITE), Pharmaceutical Technology & Development, Operations, AstraZeneca UK Limited, Macclesfield, SK10 2NA, UK

⁷ UCB S.A, 60 Allée de la Recherche, 1070 Brussels, Belgium

We would like to thank the reviewers for their interest in this work and for taking the time to thoroughly review the paper. All comments have been responded to, and the manuscript is revised accordingly.

Text Colour Key:

Reviewers Comment

Authors Response

Changes in Paper

Reviewer #1:

The application of AI, digitalization, and automation holds immense potential to accelerate the development of new drugs. Research in this vital area can significantly impact the health of patients worldwide. This paper represents an important first step in developing a self-driving tableting station. The authors have demonstrated novel contributions in three main areas:

1. Digital Formulation of Blends: This approach ensures the production of tablets with optimal quality attributes.
2. Self-Driving Tableting Station: This system can orchestrate multiple tasks to convert blends into tablets efficiently.
3. AI-Driven Control and Optimization: The use of AI model to control and optimize the tableting process.

These innovations make the manuscript worthy of publication in Nature Communications. However, before I can recommend its publication, I would like the authors to address a few key points listed below:

Comment 1 (C1). Compatibility of Excipient and API: The Digital Formulator is missing a crucial element—the compatibility of excipients with the new API. This is the first step in designing a formulation and should be addressed before focusing on other attributes like blend flowability and tablet tensile strength.

Response: We agree with the reviewer that this is a critical aspect of formulation development. The Digital Formulator addresses this by enabling the user to specify a subset of excipients for optimisation (see revised Figure 1 and 2). This functionality facilitates integration with existing digital assessment tools^{1,2}, allowing the Digital Formulator to focus on compatible API-excipient combinations.

Changes: We have revised Figures 1 and 2 and added additional explanation to the section “RESULTS - Digital Formulator: In-silico Optimisation” in the revised manuscript to highlight this functionality.

C2. Scale-Up Challenges: The main challenge in the CMC development process is scaling up from bench scale to pilot and commercial scales. Powders often exhibit physical effects like over-lubrication, segregation, and flow issues such as jamming and arching, which may not be present at small scales. Typically, early-stage industrial development is carried out with an eye towards larger scale, and appropriate scale models are developed to bridge between scales. The formulation and process optimization approach presented here may be sub-optimal at a larger scale.

Response: The current platform addresses some of these challenges by maximising flowability of the blend to minimise the risk of flow-related problems and by assessing key scale-up parameters, pre-compression pressure, main compression pressure and dwell time (multi-output Bayesian optimisation (MOBO) framework presented in Section 3.2.3 in Supporting Information), to mitigate lamination and capping risks. Flowability is implicitly considered in the optimisation process through its influence on measurable tablet attributes. The MOBO provides a framework to assess the effect of pre-compaction pressure, main compaction pressure, and dwell time on scale-dependent properties such as elastic recovery, tensile strength and porosity. Elastic recovery was chosen as one of the parameters as it gives an indication about lamination risk³. While the current version does not explicitly account for over-lubrication or segregation, these aspects are acknowledged as critical. We believe the proposed platform provides a foundation for future integration of additional scale-

dependent factors, ultimately supporting more holistic and scalable formulation and process development strategies.

Changes: We have highlighted this important point and acknowledged the limitations of the current platform in the section “DISCUSSION” in the revised manuscript.

C3. Tablet Shapes, Thickness, and Tooling: The paper and supplementary material do not mention tablet shapes, thickness, and tooling used to manufacture tablets. Model predictions of process parameters like pre-compression and main compression pressures to achieve target tablet tensile strength and porosity will depend on the tablet shapes and tooling used. This information is crucial for accurate model predictions and should be included.

Response: The model is trained for round tooling with a size of 9 mm. The thickness varies for tablets as we trained the model for compaction pressure, i.e. thickness will change in response to the compaction pressure. We used tensile strength instead of breaking force (or hardness) to make this parameter independent of the diameter and thickness. Tablet shapes is not considered in the model at this stage due to the lack of data across different shapes. We focused our efforts on consistent, high quality data for one shape to date.

Changes: Information about shape and diameter of tooling has been highlighted in section “RESULTS - Tableting DataFactory: Make & Test” in the revised manuscript.

C4. Porosity Constraint: The porosity constraint ($> 15\%$) used for in silico optimization is counterintuitive. It will end up over-constraining the space for feasible solutions. The usual industrial practice is to increase the solid fraction or reduce porosity until the desired tablet hardness or tensile strength is achieved.

Response: We used this constraint to ensure that we have sufficient porous tablet for fast liquid uptake and, consequently, effective disintegration of the tablet.⁴ The specific value of 15% is based on recommendations from cross-industry papers, particularly the Manufacturing Classification System (MCS) paper, which suggests a solid fraction of 85% (equivalent to 15% porosity).^{5,6} Lower porosities can increase the risk of capping and lamination due to the formation of localised high-density regions, while higher porosities may still yield acceptable performance. As shown in the additional disintegration data provided in Figure S41 of the Supporting Information, the selected 15% porosity consistently resulted in acceptable disintegration times across all case studies. Importantly, porosity and tensile strength constraints are input parameter for both the Digital Formulator and the Tableting DataFactory, allowing users to tailor these settings for specific formulation requirements.

Changes: We have added a short description to clarify this in the section “RESULTS - Demonstration of Platform Workflow” of the revised manuscript.

C5. NIR Method Development:

- How much effort went into developing the NIR method for assessing blend 57eneity and tablet content uniformity? This should be included in Table 2.

Response: Hotelling’s T2 analysis (Section 4.3.1 in Supporting Information) was used to monitor blend homogeneity. The collection of the spectra is fully automated and part of the Make & Test workflow (Figure 4a). There isn’t any additional effort

needed in the platform workflow for this aspect. To clarify, we did not develop a chemometric model for these use cases, even though this is possible and has been done for other ongoing studies.

Changes: We added additional information to the section "RESULTS - Tableting DataFactory: Make & Test" in the revised manuscript to clarify this.

- NIR spectra are known to depend on API particle size and blend bulk density. How was the baseline spectral drift corrected?

Response: The raw NIR spectra were first "trimmed" to exclude non-informative wavelength regions. Next, each spectrum was subjected to Standard Normal Variate (SNV) correction, which removes additive and multiplicative scatter effects and thus corrects any overall baseline offset. After SNV, a Savitzky-Golay (SG) filter was applied to smooth out random noise. Finally, the SG derivative was taken to remove any remaining sloping background and sharpen overlapping peaks. Together, these pre-processing steps reduce noise, minimise scatter-induced artifacts, correct baseline drift, and enhance the true spectral features of interest.

Changes: This description has been added to Section 4.3.1 in the Supporting Information.

- Was the NIR model recalibrated to accommodate changes in API particle size and blend bulk density?

Response: In this study we did not calibrate a chemometric model and used Hotelling T² analysis (see also response to C5). We therefore did not require re-calibration for each case study.

Minor Points:

C6. On line 238, the mass fractions of CCS and MGST have been flipped.

Response: This typo has been corrected in the manuscript.

C7. The correct Y-axis label for Figure S12(b) should be 'Tablet Porosity (%)'.

Response: This has been corrected in the revised Figure S43b (previously Figure S12b).

C8. It is unclear if section 7.4 on 'Extended Reality' in the main paper adds any scientific value. It should be moved to the supplementary information to help reduce the length of the paper.

Response: This section has been moved to the Supporting Information as suggested by the reviewer. We kept a short description in the section "RESULTS - Tableting DataFactory for Manufacturing" in the revised manuscript that refers to the Extended Reality development and application.

Addressing these points will strengthen the manuscript and provide a clearer understanding of the platform's capabilities and limitations.

We would like to thank the review and agree that these comments have indeed strengthened the manuscript.

Reviewer #2:

This study presents an integrated platform for tablet formulation and manufacturing, built around a Digital Formulator and a Self-Driving Tableting DataFactory. This is an interesting study. However, the study may be better suitable for pharmaceutical journals. Several technical issues are to be solved.

C9. The dataset consists of 113 tablet formulations with 653 data points collected under different compression pressures. The authors employ deep neural network (DNN) models to predict the porosity and tensile strength of tablets containing previously unseen active pharmaceutical ingredients (APIs). Given the small size of the dataset and the complexity of the DNN algorithm, is there a risk of overfitting?

Response: The reviewer raises a valid concern about the potential risk of overfitting when using deep neural networks (DNNs). We mitigated this concern through different safeguards during the model development and validation:

1. Significant increase of size of validation data set to assess performance of models across more distinct APIs and formulations. A summary of the data set used in revised manuscript is compared against the data set size of the original manuscript in the following table:

	Total number of data points	Total number of different formulations	Total number of different APIs (different API grades)
Original manuscript	149	16	2 (2)
Revised manuscript	546	57	11 (14)

2. The ensemble learning approach reduces overfitting by averaging out model-specific biases and provides uncertainty estimates through the standard deviation of predictions.
3. A leave-API-out validation strategy was used to evaluate generalisation performance to previously unseen APIs, which provides a more stringent test of model robustness than conventional random splitting.
4. The DNN architecture was kept simple with only two hidden layers of 128 units each, following ReLU activation functions, to limit model complexity relative to the available data and balances model expressiveness with the risk of overfitting.
5. The superior performance of DNNs over Random Forest and Support Vector Regression in the leave-API-out validation suggests that the ensemble DNN approach successfully captures generalisable patterns rather than memorising training-specific features.

Changes: Additional information about size of training and validation data set has been added to the section "Digital Formulator: Hybrid System of Models" in the revised manuscript. Additional explanation about the assessment of over- and underfitting has been added to the section "METHODS - Digital Formulator: Hybrid System of Models" in the revised manuscript and Figures S10 and S11 in the Supporting Information.

C10. The study only contains several drugs and excipients (e.g. MCC, lactose, CCS, MgS). Although these excipients are widely used in tablet formulations, there are much more other excipients in tablet formulations.

Response: The reviewer correctly notes that the study includes a focused set of excipients compared to the broader pharmaceutical landscape. The selected excipients

represent the core materials commonly used in direct compression formulations, ensuring the modelling framework is particularly relevant and applicable for these materials. We restructured several sections to more clearly communicate the number of materials used in various parts of this study. For the development of the Digital Formulator (training and validation) we used a total of 170 different formulations that were comprised of 15 different APIs (18 different API grades) and six different filler/binder grades. We added more information about the excipient library which includes 32 grades of excipients including various microcrystalline cellulose, mannitol, lactose, dicalcium phosphate anhydrous grades. The demonstration of the platform workflow was then conducted on a subset of six filler/binder grades (Figure 5a in revised manuscript). It is also important to note that the modelling framework is designed to be material-agnostic, using exclusively fundamental material characteristics (bulk density, true density, particle size and shape distributions) rather than material identity, enabling generalisation to new materials provided their properties fall within the property space used to train the models (Figures S1 to S5 in the Supporting Information).

Changes: Details have been added to the “RESULTS - Digital Formulator: Hybrid System of Models” and Figure 5a in the revised manuscript. More details about the materials have also been provided in Tables S1, S2 and S3 and Figures S1 to S5 in the Supporting Information.

C11. How do the small-scale formulations transition to large-scale manufacturing batches?

Response: We implemented a scale-up assessment using the Tableting DataFactory (see response to C2). However, the direct comparison to commercial-scale manufacturing batches was not feasible due to the limited access to commercial-scale lines or associated published raw material, process and product data. While several publications report data from commercial-scale lines, these typically involved proprietary APIs^{7,8} to which we do not have access. Nonetheless, we recognise the importance of such comparison and are actively pursuing opportunities to compare our small-scale development outputs with commercial-scale manufacturing data in future work.

Reviewer #3:

Authors report a detailed description of an approach for intensifying powder based product design with an example from pharmaceutical products. This piece of work is elegant engineering, programming and automation example, ultimately orchestrating powder handling flow in a (potentially) self-optimizing manner.

C12. However, the reported work remains very ‘developmental’ focusing on increasing the number of experiments that can be performed, with a limited focus on scientific questions! Current manuscript is focusing on an approach simply aiming to find a (local) optimum composition without asking the fundamental question ‘Why?’. Authors do not really go into using this approach to understand the physics of particulate systems better. What are the underlying reasons for powder flow, compaction of particles, strength of a compact – what is affecting charging of particles (particle surface) -> flow, what is the effect of particle size distribution, why does a compacted powder form a solid compact with a high enough tensile strength?

Please discuss this properly and strengthen this part in the manuscript.

Response: The reviewer raises an important point about the need for fundamental scientific understanding alongside technological development. While we respectfully disagree that the work is "developmental", we acknowledge that the original manuscript could better emphasise the scientific insights gained from this integrated approach. The reviewer's concern about moving beyond optimisation to understand the "why" behind particulate system behaviour is well-taken and represents a valuable opportunity to strengthen the scientific contribution. To address the reviewer's concern about fundamental scientific understanding, we have conducted an additional systematic investigation examining the influence of active pharmaceutical ingredient (API) particle size distributions (PSDs) on flowability and tensile strength across various drug loadings. Moreover, a sensitivity analysis is conducted to explore the variation of excipient concentration ratio in all optimised formulations achieved by in-silico optimisation and Digital Formulator to gain insights into the influence on the excipients on the decision making.

Changes: We have added new sections “RESULTS - Impact of API particle size distribution on manufacturability criteria”, “RESULTS - Sensitivity analysis of optimised formulations” and “RESULTS - Feature importance analysis” to the revised manuscript and new Sections 4.2.2 and 4.2.3 in the Supporting Information.

C13. The second major comment is the target product profile. What about the most critical property of a compacted pharmaceutical tablet for immediate release, disintegration? Authors do not mention disintegration nor measure any disintegration behavior (or very shortly, mainly pointing to ‘future studies should...’). It can be assumed that tablet porosity correlates with disintegration time, but this is quite far fetch with the current materials.

Please consider including this into the work properly.

Response: We agree that disintegration is a critical property, and its inclusion strengthens the link between tablet porosity and in-vitro performance. We therefore have performed disintegration testing on tablets with the desired porosity across the nine case studies. Across all case studies, the tablets satisfied the requirement of disintegrating in less than 15 minutes (Figure S41 in Supporting Information), in line with expectations for immediate release tablets.

Changes: We have added additional details to the section “RESULTS - Demonstration of Platform Workflow”, and have provided the disintegration data in Figure S41 in the Supporting Information.

C14. Third major comment: state of the art on self-optimizing labs is dominantly with liquid phase system (organic reaction optimization in flow reactors etc). Authors have a point in their work, and they have a clear novelty, because now they operate with powders, solid matter. This important aspect is not really spelled out for the reader. Please compare and contrast properly this aspect of novelty in the Introduction chapter of the work.

Response: We agree with the reviewer that this aspect has not been highlighted clearly in the introduction. We have revised the Introduction to explicitly compare and contrast the current state-of-the-art in self-driving laboratories, primarily focused on liquid-phase systems, with our novel approach involving powders and solid matter. This clarification emphasises the unique challenges and contributions of our work within the broader context of autonomous lab systems. The new paragraph has been highlighted in Introduction section of revised manuscript.

Changes: We have added a paragraph to the section “INTRODUCTION” in the revised manuscript.

C15. Fourth one: authors refer to physics-based models. This is a strong expression for basically curve-fitting (empirical models of Kawakita and R-D). Here it is suggested that authors tune down the physics, especially considering that they are not really discussing the fundamental physics of powder flow and underarticulate interactions affecting tablet processes.

Response: The reviewer raises a valid point regarding our use of "physics-based models" to describe the Kawakita and Ryshkewitch-Duckworth (R-D) models, which are fundamentally empirical relationships. We revised the terminology throughout the manuscript to “empirical models”, which better captures that these models incorporate physically meaningful parameters and relationships while acknowledging their empirical foundation. Despite their empirical nature, these models remain valuable components of the hybrid framework as they capture physical relationships. They provide structure and physical constraints that guide the data-driven components of our system.

Changes: We have revised the use of terminology from physics-based models to empirical models when referring to the Kawakita and Ryshkewitch-Duckworth (R-D) models.

More minor comments, but still something that should be addressed:

C16. The chemical space of both excipients and active pharmaceutical ingredients is quite limited. Basically few excipients (cellulose, lactose, calcium phosphate and mannitol) and six/eight simple APIs. Also, considering the chemical space: authors have an interface to Cambridge Structural Database (CSD). However, very few descriptors are used for eight materials. Considering that CCDC is advertising on their homepage that they have 1.25M accurate 3D structures, the author’s discussion seems very limited.

Response: The selection of crystallographic descriptors from the Cambridge Structural Database was deliberately limited to prevent overfitting. The reviewer's reference to 1.25M structures refers to different structures/materials in the database,

not descriptors - we systematically adopted the relevant crystallographic descriptors for every API studied in this work.

The crystallographic descriptors showed only marginal improvement in prediction accuracy compared to particle-based descriptors, which is mechanistically justified by the scale mismatch between molecular-level crystal structure and tablet mechanical properties governed by particle-level interactions. In pharmaceutical tablets, APIs are dispersed within excipient matrices where particle-particle interactions and excipient mechanical properties dominate overall tablet behaviour, diluting crystal structure effects.

The deliberate limitation of CSD descriptors reflects common machine learning practice for moderate datasets, preventing overfitting while prioritising model robustness. The marginal contribution validates that tablet mechanical properties are primarily governed by particle-level phenomena rather than molecular-level crystal structure, suggesting formulation scientists should focus on optimising particle properties and processing conditions before crystal form modifications for mechanical property enhancement.

C17. Considering powder flow and tablet compaction, it is well-known that particulate properties are of critical importance. As a continuation of the previous comment on ‘chemical space’, the ‘particulate space’ is quite limited: APIs are relatively large particle size and no really challenging APIs are included in the experimental data set. Please comment on this and revise accordingly.

Response: The reviewer raises an important point about the particulate space in our dataset. The expanded list of API grades used in this study includes materials with a d₅₀ of 20 – 30 µm (Ibuprofen 25, Dexamethosone and Griseofulvin). Dexamethosone and Griseofulvin were also used as case studies to demonstrate the platform workflow. 14 materials used in the study are also classified as cohesive and very cohesive materials (FFC < 4) which is a good representation of industry relevant properties. However, we acknowledge that the parameter space could be further expanded to include materials with more challenging particulate properties (such as micronized APIs).

As highlighted in our response to the Comment C10, the modelling framework is designed to be material-agnostic, training on fundamental particulate descriptors rather than material identity. The models do not differentiate between excipients and APIs, instead focusing on particle size distributions, density, flow properties, and morphological characteristics. This approach enables the framework to accommodate materials with different particulate properties, including challenging APIs, provided their characteristics fall within the expanded property space of future datasets. The hybrid modelling system's ability to predict tablet properties across different drug loadings demonstrates that the framework can handle varying contributions of API particulate properties to overall formulation behaviour.

Changes: We have added this important point to the section “DISCUSSION” in the revised manuscript.

C18. Strengthen the description of particulate space, and discussion related to this.

Response: To address this concern, we have conducted an additional systematic investigation examining the influence of API particle size distributions on tablet

mechanical properties across various drug loadings, which provides deeper insight into the particulate space coverage and its impact on formulation behaviour.

Changes: We have added a new section “RESULTS - Impact of API particle size distribution on manufacturability criteria” to the revised manuscript.

C19. Same Table S1: textbook is teaching us that MCC Avicel PH101 is closer to 50 microns and PH102 is 100 microns. Here the values are opposite. This is most likely a typo, BUT raising a concern of the overall quality of the raw data. It is appreciated that data is available in the Source Repository, but readers should be able to trust the data in the times when everything is easy to make automated and just copy-paste the errors to the next generation of scientists.

Response: We thank the reviewer to point this out. We reviewed all the data in the tables and corrected any errors. This did not affect any results as it the error only occurred in the manuscript and not the raw data that was used for the modelling.

Changes: Table S2 (previously S1) has been updated with correct information.

C20. Authors make it difficult to capture the composition of their model system. Line 239 is stating that there is 3.5 w% of MgSt (a really high amount), but Supporting info is saying 1 w%.

Response: This typo has been corrected and highlighted in section “RESULTS - Demonstration of Platform Workflow”.

C21. Supporting info Line 28: formulated as presented in Table S1?? Please proofread your text.

Response: This has been corrected in Supporting Information.

C22. Throughout the text; authors refer to Figures in the main text and supporting information randomly without always indicating S for Figure in the supporting information. Please proofread the text carefully for that aspect.

Response: We have proofread the manuscript and revised all figure references to clearly distinguish between main text figures and those in the Supporting Information by consistently using the "S" prefix where appropriate.

C23. Table 2 in Conclusions seems to appear in a wrong place. It can be included in one of the earlier chapter as a discussion element, but it seems wrongly placed when in Conclusions.

Response: We appreciate the reviewer’s observation regarding the placement of Table 2 in the Conclusions section. We have decided to convert the table into text and have integrated the relevant information into the RESULTS section. We believe this improves the flow of the manuscript.

Change: We converted the table into text in the section “RESULTS - Benchmarking and Performance Metrics” in the revised manuscript.

C24. Minor minor comment: the use of abbreviation XR might confuse a reader with pharmaceutical background. This abbreviation is often use for extended release (XR) products. Here authors report work on immediate release. Otherwise, very elegant chapter and novel approach within the pharmaceutical community.

Response: Thank you for pointing this out. The abbreviation "XR" has been replaced with its full form, *extended reality*, throughout the manuscript.

Change: Most of the extended reality-related content has been moved to Section 5 of the Supporting Information for clarity and focus.

Review #5:

Reviewer report for Nature Communications manuscript #NCOMMS-25-15811 Manuscript title: Accelerated Medicines Development using a Digital Formulator and a Self-Driving Tableting DataFactory.

C25. In this study, the Authors discussed an automated tableting platform that leverages machine learning, modelling tools such as physics-informed Bayesian optimization (PIBO), and robotic automation. The authors offered a material-to-product approach to predict and optimise critical quality attributes for different formulations, linking raw material attributes to tablet properties such as flowability, porosity, and tensile strength. The motivation for the work is important, namely to improve the sustainability and efficiency of direct compaction tablet drug product development. Substantial research and development efforts are noted and applauded for. However, the paper does not represent significant enough scientific innovation. The paper employed mostly established, or the author group's previously established methods, and did not report any scientifically significant findings.

Response: We respectfully disagree with the reviewer's assessment regarding the level of scientific innovation presented in this work. We believe our study introduces two key innovations that, when integrated into a unified platform, represent a significant advancement in the field of pharmaceutical product development:

- **Digital Formulator:** To the best of our knowledge, this is the first model capable of predicting critical tablet properties, specifically porosity and tensile strength, directly from raw material attributes, without requiring prior blending or tablet testing for model calibration. While the Digital Formulator integrates concepts from established mixture models and Particle Informatics, it introduces several novel elements: new process models, an in-silico optimisation engine, and a curated excipient library. Together, these enable in-silico formulation development, allowing the selection and optimisation of excipient combinations and their respective mass fractions under defined constraints and target properties. This capability goes beyond prior work and offers a transformative approach to early-stage formulation development.
- **Self-driving Tableting DataFactory:** This is the first reported system capable of automating the tablet development workflow, from powder dosing and transport, through compaction, to testing, on a per-tablet basis. While self-driving laboratories are increasingly reported, they have predominantly focused on liquid handling. Our work extends the self-driving concept to the complex domain of powder-based systems, which introduces unique challenges in material flow, compaction control, and mechanical testing.

Given the lack of comparable platforms, either for each individual component or for the integrated combination of predictive in silico formulation and automated workflows in tablet development, we believe our work represents a significant advancement in both innovation and practical impact.

To further support this point, we have added new sections to highlight scientific findings that were uniquely enabled by our platform. These include new insights into formulation-process-property relationships.

Changes: We have restructured the final paragraphs of the introduction section in the revised manuscript to better highlight the key elements of the platform. We also introduced the new sections "RESULTS - Impact of API particle size distribution on manufacturability criteria", "RESULTS - Sensitivity analysis of optimised

formulations” and “RESULTS - Feature importance analysis” to discuss key scientific findings.

C26. While the time and material saving claims are noted, they lacked substantiation. The material and time saving by itself is considered as lack of scientific significance by the reviewer. Automation is naturally expected to speed things up as an incremental engineering advancement rather than scientific innovation. In addition, the writing of the manuscript is significantly below the standards of Nature Communications. The discussion is confusing to such an extent that the scientific quality of data is called into question.

Response: The platform is primarily designed for early-stage pharmaceutical formulation and process development, where API availability is often severely limited, and flexibility and rapid iteration are critical. In this context, our focus is on minimising material usage while maximising the knowledge generated for a given API.

At this stage, it is challenging to ensure fully equivalent benchmarking across all components of our platform, particularly when comparing to manual laboratory workflows that lack integrated in-line monitoring and automated testing and data logging. Rather than attempting to force equivalence where it does not exist, we have chosen to highlight the distinct capabilities of our system, namely its integrated automation and data generation features.

Given the constraints of early development, we placed particular emphasis on demonstrating material savings, as detailed in our response to Comment C57. Time savings are also reported; however, these are more difficult to benchmark precisely, as they are not quantified in the literature. Our estimates are based on the industrial experience of the collaborators involved in this work.

Beyond material and time efficiencies, this platform contributes to generating new understanding across material, blend and tablet attributes. The combination of the Tableting DataFactory producing high-quality, structured data that feeds into the Digital Formulator enables systematic knowledge generation and supports informed decision-making in formulation development. We have added examples of the new knowledge generated in the revised RESULTS section.

Change: We have added a new “RESULTS - Benchmarking and Performance Metrics” in the revised manuscript. We have reviewed the entire manuscript and have revised several sections to improve readability and quality of the manuscript. We have moved several aspects into the Supporting Information to make it easier to follow the manuscript.

The reviewer provides the following review feedback with the intention of being as constructive as possible. The reviewer recommends the authors, after major revisions, to re-submit the manuscript to a more process engineering-oriented journal, potentially as 2 manuscripts. The reviewer recommends rejection of the manuscript from further publication consideration by Nature Communications.

Major concerns on Figures and Tables: Setting aside for a moment the content, the presentation clarity and quality of the majority of the figures and tables do not meet Nature Communications standards.

C27. Figure 1 was intended to clarify the overarching workflow. Instead, the reviewer found it to cause more confusion. Please clearly annotate and explain:

1. The relationship among the blocks.
2. Which blocks represent measurements, and which represent prediction.
3. The relationship between shaded blocks and bullet texts without shade.
4. The meaning of different greyscale shades of the shaded blocks, if any.
5. The blocks under thick arrow – are they supposed to be the input or output to the main block (e.g., “Material Characterisation” – shouldn’t particle size be an output rather than an input?). If the arrows do not imply input, output or directionality, it should not be utilized.

Response: We considered the reviewers comments and updated Figures 1, 2 and 3 as well as improved consistency in the terminology used in the main text and the figures.

Changes: *We have updated Figure 1 in the revised manuscript and updated terminology in the main text.*

C28. Figure 5 was referenced (line 308) but was not annotated in the figure.

Response: This has been corrected.

C29. Why did “Tableting DataFactory” show up twice? Does Material Characterization only input to Digital Formulator, but not to Tableting DataFactory?

It is disappointing to the reviewer that the authors started a Nature Communications manuscript Figure with a pile of bullet points, indicating a lack of architectural deliberation, presentation clarity, or both.

Response: We changed this figure to improve clarity.

Change: *We have updated Figure 1 to improve clarity of the schematic illustration.*

C30. Figure 2 is an expansion of “Digital Formulator” from Figure 1.

- “Hybrid system of models” block is consistent with Figure 1, but “In-Silico Optimisation” is not. Optimization appeared several times in Figure 1, thus it is impossible to correspond and rationalize how these optimization modules in Figure 2 relate to that of Figure 1.
- Particle size appeared again several times in “Material (Powder) Space”, presumably as the input of “Mixture models”. How are they related to the “Material Characterisation” module in Figure 1?
- Similar question to Figure 1, what properties are measured, literature/database retrieved, or predicted?
- Is “New API characterization” part of the optimization loop, as the current figure suggests?

Response: We created a new figure to address the comments and improve clarity.

Changes: *We have added a revised Figure 2 to the revised manuscript.*

C31. Figure 3 – Is prediction accuracy only demonstrated on training data? If so, validation predictions on parameter space that were not part of the training should be shown to allow the reviewer to truly assess the model’s predictability. If the validation predictions are already in the plot (16 out of 113 as suggested in the method section), they should be clearly differentiated from the data used in the training. In Figure 3b, there is a range on the left of the histogram,

>2.7MPa, that does not have any data. The corresponding range in the cross-correlation plot below showed a lot of data. Please explain. Why are the histograms in Figure 3 and Figure S1 different?

Response: The reviewer's concern about training/validation data differentiation is valid and represents a critical aspect of model evaluation that requires clarification. We acknowledge that the original presentation did not adequately communicate between training and validation data. We have now revised the text to clearly communicate this, and expanded our validation approach. The original dataset comprised 116 formulations (653 data points) for training and 16 formulations (149 data points) for validation, but we have since increased the validation dataset size to provide more robust assessment of model generalisability (the updated validation dataset include 57 formulations with 546 data points). We have added further explanation to the response to the related Comment C9.

Changes: *The updated manuscript now explicitly differentiates training and validation data throughout all figures and analysis. A summary of the data has been added to section "RESULT - Digital Formulator: Hybrid System of Models" in the revised manuscript with further details in Section 2.1 in the Supporting Information.*

C32. Figure 4 – What is the purpose of rows with all zero values? Discuss or remove.

Response: Summary of results [DM1] from formulation optimisation cases. The x-axis represents different case studies with target APIs and their corresponding mass fractions (% w/w). The y-axis lists the subset of excipients considered in the formulation optimisation process. The colour scale indicates the mass fraction (w/w) of each excipient. The '0.0' indicates that this excipient was not chosen by the Digital Formulator for the given use case.

Change: *We have updated Figure 5 (previously Figure 4) and its caption in the revised manuscript.*

C33. Figure 5 needs major improvements. The robotic arm icons are very small and difficult to distinguish between #1 and #2. Consider bright colors such as Red and Blue that can help. The flow chart in panel (a) is difficult to decipher; the light gray arrows should have darker gray colors. In panel (b) some of the text is going outside the textbox, which is unacceptable. How is Figure 5 related to Figures 1 and 2? Consistent terminology and colour coding of blocks are essential to present a complex, layered workflow. SCU is referenced repeatedly in the text, but not annotated in the Figure. Modules are mentioned without any sequence, nor annotation, making the narration extremely difficult to follow.

Response: This figure has been updated according to the comments of the reviewer. We also replaced SCU with orchestration system across the entire manuscript.

Changes: *We have updated Figure 4 in the revised manuscript and updated terminology in the main text.*

C34. Figure 7 - This figure is overall poorly discussed. The caption referred to two tuning parameters as a and b, while there were no references of the four sub-figures a, b, c, and d. The figure showed a lot of data, however, was only briefly explained in line 402, which neither helped clarify the main point of the figure, nor offered explanation of the data. A plateau was referenced but was far from universal. Some clearly did not reach a plateau (Using Figure 7a as an example, red, orange, pink, and green did not reach a plateau). Why? Termination criteria were briefly discussed, but produced confusing results – why is red clearly not plateaued, but

terminated, while grey, brown and green are plateaued in iterations #3 and #4 already, but not terminated and ran to iteration #5? Why does the parameters remain zero for both iterations #1 and #2, and for all the parameters? They cannot be simply explained as “underestimation of the initial predictions”. Rather, some systemic initialization issue is called into concern.

Response: The figure caption identifies the subfigures (a: ϵ_0 , b: B, c: \hat{T} , d: kb), with each subfigure displaying the trend of the respective parameter from the models. Model fitting was initiated after iteration 2 to avoid unreliable parameter estimates from using only two data points. Consequently, the PIBO algorithm is programmed to initiate tuning parameter monitoring from iteration 3 onwards, when sufficient data becomes available for meaningful model fitting.

The main purpose of this figure has been to demonstrate the convergence of tuning parameters (which shows the successful fit to the empirical model). It is worth clarifying that the termination of PIBO is based on the change in error between tuning parameters and the data, as shown in Eqs. 4 and 5 in the revised manuscript. While the acquisition function is constrained to reduce the error between collected data and empirical models, in some cases such as SP 22%, the constraints might be violated for some iterations. This can be attributed to the inherent noise in data that lower the quality of fit, even though that one tuning parameter does not change significantly. While cases such as SP 16%, SP 22%, and GR 20% terminated at iteration 4, others including MH 20% and IM 20% continued for additional iterations. All formulations achieved the convergence criterion of less than 20% change in the Err_n as compared to Err_{n-1} , indicating stabilisation of the empirical model parameters. Clarifying text and mathematical notations are added to the manuscript to explain both the rationale for initial parameter values in early iterations and the impact of error and data noise on parameter convergence and target value achievement in the optimisation process.

Change: We have restructured the manuscript to bring together the results and discussion of the nine case studies. We combined Figures 4 and 7 and Table 1 into the new Figure 5 to improve readability of the manuscript and clarify the points raised in this comment.

C35. Figure 8 is insufficiently discussed.

Response: We restructured the manuscript to keep the focus on the key innovative elements. We moved this figure and discussion to the Supporting Information. We believe the restructured manuscript and Supporting information improve the overall readability and, more specifically, the readability and discussion of this figure.

Change: We have moved this section and figure to the Supporting Information.

C36. Figure S2 is considered by the reviewer as one of the primary scientific findings from this manuscript. It is discussed extensively in the main manuscript body by the authors, while the figure is included as Supporting Information. The inconsistency is an ignorance at best. It should be included into the main manuscript. The discrepancy is substantial, and the explanation is insufficient. 2 out of 9 data points failed the FFC requirements, 4 under predicting, 5 over predicting, and some with error over 50%. This should be not claimed as “This consistency”.

Response: We appreciate the reviewer’s emphasis on the scientific importance of Figure S2. Flowability is normally interpreted in categories – cohesive ($FFC < 4$), easy flowing ($4 \leq FFC < 10$), and free-flowing ($FFC \geq 10$). The model correctly placed 6 out of 9 formulations in the appropriate class, thereby preventing critical misclassification. This model performance is close to the original study on the development and validation

of here-employed data-driven mixture model.⁹ Also, no flowability issues were observed during the process of all formulations, including those that were misclassified. It is worth noting that most regression models applied in this context will inevitably show both under- and over-estimation; what matters is whether those deviations are acceptable for decision-making.

For powders with higher FFC values, absolute flow issues are negligible, so even larger percentage errors do not affect practical handling recommendations. Although individual FFC predictions show both under- and over-estimation, the model accurately guided the filler/binder optimisation for successful powder handling during the tableting process, leading to consistent weight uniformity across the different case studies.

While there are inherent errors in model predictions, we emphasise the model's superior performance as compared to the state-of-the-art using regression¹⁰ (Yanes, et al. ¹⁰ recently developed a regression model and reported the R^2 of 0.78) and classification¹¹ (Owasit, et al. ¹¹ developed a classification model and reported the proportion of correct predictions relative to the total number of instances, yielding accuracies of 40% for the individual dataset and 42% for the blends) approaches. This can be compared with the overall accuracy of the FFC model utilised in this work, which showed the R^2 of 0.93 through the validation based on 44 materials and 220 datapoints.

Change: We have added revised text in the section "RESULTS - Demonstration of Platform Workflow" to clarify these points, as well as an additional validation figure (Supplementary Figure S6) of the FFC model with the latest data in the supplementary information to support the prediction performance of the model.

C37. Table 2 should not be in the conclusion section. It needs to be completely redesigned for a journal publication. Table should focus on presenting the numbers, while the text should be incorporated into discussions. All numbers need to be substantiated with data from the paper or literature. For example, it is very difficult to believe the state of the art method is "manual filling" or "shoe feeder". Shouldn't an automated system intended for Nature Communications be compared with a state of art system used in the industry? If it is compared to a laboratory specific manual filling method, then the benchmark is not state of art, and the claims on wider industrial impact need to be revised. Further, to follow the author's claims that the 72 tablets per hour rate included powder quality monitoring, tablet manufacturing, testing, storage and data handling, should the corresponding time using manual filling include those monitoring, testing, and handling elements as well? The table stated 72 tablets per hour, while the introductory section emphasized 1440 tablets in 24 hours, which is 60 tablet per hour. It is again inconsistency and ignorance at best.

Response: We thank the reviewer for the detailed feedback regarding Table 2 and the benchmarking approach. We acknowledge that the table, as originally presented, was overly qualitative and placed inappropriately within the conclusion. In the revised manuscript, we have removed Table 2 from the conclusion and redesigned it to focus on presenting clear, quantitative performance metrics. This information has been moved to the RESULTS section in the revised manuscript.

Manual filling and shoe-feeder (and sometimes paddle feeder) setups remain the de facto standard practice in many academic and industrial R&D laboratories. Research studies and formulation development protocols still rely on manual operation at this scale.¹²⁻¹⁴ Hence, we consider these methods to represent the *state-of-the-art* in the context of our study.

Industrial tablet manufacturing systems indeed achieve significantly higher throughputs. However, they are optimised for large-scale production and are not suitable for small-batch, material-sparing development workflows. Our work fills this gap by providing a self-driving, end-to-end automated platform tailored for research and development use, and thus, our benchmarking focuses on methods used in this phase rather than full-scale manufacturing equipment.

We also appreciate the reviewer highlighting the inconsistency between the 72 tablets/hour and 1,440 tablets/day figures. To clarify: the system's maximum operating throughput is 72 tablets per hour, considering all powder doses and tablets meet the requirements (decisions D1 and D2 in Figure 4 in revised manuscript). The daily capacity of 1,440 tablets over 24 hours represents a more realistic operational estimate, accounting for periodic interruptions such as cleaning cycles, hopper refilling, or brief maintenance, thus averaging 60 tablets/hour. We have now explicitly clarified this distinction in the manuscript to avoid any confusion.

Regarding the process stages: we appreciate the suggestion to ensure throughput comparisons include equivalent steps such as powder quality monitoring, testing, data handling, and storage. However, at this stage, it is challenging to include fully equivalent stages as pointed out in our response to Comment C26. Our focus remains on demonstrating the system's integrated capability.

Change: Table 2 has been removed from the conclusion and its contents have been converted into text, which is now incorporated into the section "RESULTS - Benchmarking and Performance Metrics". Clarification about number of tablets per hour and per day is also added to this section.

Fundamental Modelling Concerns

C38. Is 113 tablet formulations (653 data points) considered sufficient? The reviewer has doubt on the massive parameter space that can be representatively covered by 653 data points.

Response: We substantially expanded the validation dataset to 546 data points to more rigorously assess the predictive performance of the Digital Formulator across a broader range of unseen materials. The updated validation includes 18 different API grades, thereby enhancing the diversity of the dataset. Importantly, model performance remained very good, comparing the previous validation data set (porosity: $R^2 = 0.95$, $RMSE = 0.02$; tensile strength: $R^2 = 0.90$, $RMSE = 0.25$ MPa) to the updated validation data set (porosity: $R^2 = 0.90$, $RMSE = 0.01$; tensile strength: $R^2 = 0.89$, $RMSE = 0.40$ MPa). This demonstrates that the Digital Formulator effectively captures the defined parameter space and accurately models the relationship between raw material attributes and blend and tablet properties.

Change: We have updated the validation data points in the revised manuscript.

C39. Can the authors comment how the 653 training data points cover the parameter space of the properties of the 9 formulations predicted? This comment links back to the training and validation comment for Figure 3. The reviewer appreciates the rich set of additional data included in "Support Information". However, the reader should be able to understand the main points supported by the main manuscript. The supplementary document should be considered as optional, not mandatory.

Response: We have added additional information to clarify the training and the extended validation data used in the study. The performance of model per each validation API has been investigated as well. To clarify, the training and validation of

the Digital Formulator was performed independently from the demonstration of the platform workflow across these nine case studies. We have restructured the revised manuscript to improve clarity and enhance the overall flow of the narrative. We provided further explanation to this in our responses to the Comments C9 and C38.

Change: We have restructured the manuscript to improve readability, and separated the training/validation of the Digital Formulator and the nine case studies.

C40. What are importance scores of the input parameters? It would simplify the computation if only the relevant input parameters are retained in the modelling.

Response: We have performed a feature importance analysis (SHAP) and added a discussion to the revised manuscript.

Changes: We have added a new section “RESULTS - Feature importance analysis” to the revised manuscript to address this comment.

C41. An ensemble of Deep Neural Network (DNN), Random Forest and Support Vector Machine (SVM) were compared. However, the claim about DNN being the “superior” model is questionable. Yes, the SSE and R2 values in Table S3 favor DNN slightly. However, DNN tends to overfit and is less reliable in the context of the limited training data. How does the limited training set (only 653 data points to train a DNN for 21 features) impact the different models differently. For example, in Figure S1 tensile panel, there are 3 outliers near 3.2 MPa which does not exist for RF.

Response: This has been addressed by multiple measurements in the revision:

Our DNN ensemble approach specifically addresses overfitting/underfitting concerns through multiple safeguards that we have demonstrated. First, we employed an ensemble of 20 DNN models with different random initialisations, which inherently reduces overfitting by averaging out model-specific biases and provides uncertainty quantification through prediction variance. Second, we implemented rigorous leave-API-out validation, which represents a more stringent test of generalisation than standard random splitting since it evaluates performance on completely unseen APIs. Third, we have provided additional validation data beyond the initial validation dataset and demonstrate convergent average training and validation loss curves over the ensemble of models, which directly contradict overfitting behaviour where training and validation performance would diverge significantly.

Regarding the specific outliers mentioned in Figure S1, these represent legitimate experimental data points that the RF captures more accurately than DNN ensemble, however, the model performance is assessed over the general ability to capture all validation data over the provided range, which actually demonstrates the DNN's superior capacity to learn complex non-linear relationships in all available data rather than evidence of overfitting or wrong prediction for only few data points. The performance metrics (e.g. R^2 , RMSE) remain the main standard for regression model evaluation and provide objective quantitative assessment of model accuracy. These metrics, combined with our validation approach, demonstrate that the DNN ensemble achieves superior predictive performance compared to Random Forest and Support Vector Machine approaches.

Changes: We have added extra clarification on the safeguards to avoid overfitting in process models, while extending the validation dataset and updating the performance metrics.

C42. More fundamentally as a study design concern, if predictions were made using full set of input parameters including calculated particle informatics descriptors (Figure 3), DNN's "superior performance" needs to be reevaluated against other models using the additional particle informatics descriptors.

Response: We employed a systematic two-stage evaluation approach to ensure fair model comparison. First, we compared DNN, Random Forest, and Support Vector Machine models using the baseline feature set without particle informatics descriptors, demonstrating DNN ensemble superiority through leave-API-out cross-validation. Second, we evaluated the superior DNN model's performance with and without particle informatics descriptors, which showed marginal but consistent improvement in reducing prediction outliers and uncertainty.

This methodology separates model architecture performance from feature engineering effects, preventing confounding factors that could arise from simultaneous comparison of different models with different feature sets. The approach confirms that DNN superiority is not dependent on additional particle informatics features, while transparently demonstrating their incremental benefits for prediction accuracy.

C43. Lines 223 onward lacks definitions on

- o Ensemble learning
- o Ensemble of models
- o Exploration
- o Exploitation and how it is different from exploration

The reviewer realizes some of this may be due to Nature Communication's requirement on moving method section after conclusion. It is the authors' responsibility to ensure narration coherency.

Response: We restructured several sections in the manuscript to improve readability and the narrative.

C44 Line 232: Is the sub-set of excipients used for the training, but not the rest of excipients? If it is the case, the other excipients are not relevant and should be excluded from the manuscript entirely. If it is not, then the reviewer has further concerns on the authors' choice in what data was included in the training, what was included in validation, and what was included for the optimization.

Response: We acknowledge that this was not sufficiently clearly described in the manuscript. We used all materials (APIs and excipients) listed in the manuscript. The majority of excipients were used to train and validate the Digital Formulator. A subset of the excipient library was used for the nine case studies. We believe the restructuring (separation of the Digital Formulator training/validation from the case studies), the revised Figures and the additional information provided in the revised manuscript clarifies this now.

Change: We have restructured the manuscript, updated Figures 1, 2 and 5 and added additional information to the "RESULTS - Digital Formulator: Hybrid System of Models" and "RESULTS - Demonstration of Platform Workflow" in the revised manuscript to clarify this.

C45 Line 235: Does the expanded list of excipients need to be included in the training?

Response: The model inputs for Digital Formulator are based on material properties and it predicts the blend and tablet attributes based on their raw material properties. We added further clarification to our response to the related Comment C10.

Change: We have updated section “RESULTS - Digital Formulator: Hybrid System of Models” in the revised manuscript to clarify the used materials and also the fundamental set up of the modelling framework.

C46 Line 249: MCC1 did not have FCC data (Table S1). How does that impact MCC1 being chosen as an excipient?

Response: We do not require the FCC data for the materials. We only need true density, bulk density, particle size/shape distribution data. We provided some additional common material properties to show the property space of these materials. However, we performed the missing measurement and have added the data for completeness.

Changes: We have added the FCC data of MCC1 to Table S2 (previously Table S1).

C47. Line 373: Porosity and tensile strength have been presented as primary prediction parameters. Why is elastic recovery introduced later? Either make it consistently part of the prediction parameters or elaborate on its delayed inclusion.

Response: The elastic recovery is a parameter to be monitored and optimised during the MOBO optimisation, it is considered as an extra target in addition to porosity and tensile strength to optimise main/pre-compression pressure and dwell time, although they are not considered in the Digital Formulator. We restructured this section and do not discuss the MOBO in the main manuscript to avoid any confusion and improve readability of the manuscript. We added additional details to the Supporting Information.

Change: We have moved this section to the new Section 3.2.3 in the Supporting Information.

C48. Line 463: GP is not defined. Figure S10 incorrectly referenced.

Response: This has been corrected.

C49. MR/AR section: This section is both poorly written and poorly integrated. The argument of its innovation and demonstration of its value are both weak. While the reviewer appreciates it as an interesting technology demonstration, it does not belong to this manuscript in general and is very far away from Nature Communications standards.

Response: This section was moved to the Supporting Information to improve readability of the main manuscript.

Grammar, language and consistency issues. These issues are throughout the manuscript with examples noted below,

C50. In the Abstract: “This approach accelerates... tablet within 6 hours...” should be “tablet TO within 6 hours.”

Response: This sentence has been revised.

C51. On page 26 footnote: “The list of informatics descriptors IS”

Response: This has been corrected.

C52. Figure 5: To reduce clutter, the abbreviations PIBO and SuMoBO should be deleted. Keep capitalization consistent (“Orchestration” versus “transfer tablets.”)

Response: This figure has been updated.

C53. Figure 6: Typo in panel (a) “DataFacotry”

Response: We removed this figure to reduce the length of the manuscript.

C54. In Table 2, “hoper” should be hopper.

Response: We have corrected this typo.

C55. There are typos in Figure S1, “SVR” should be SVM.

Response: This has been corrected.

C56. “Table 1” and “Table 1 in Supporting Information” should be better distinguished with Table S1. Similarly for other references to Tables and Figures.

Response: We have resolved this by clearly distinguishing between the main manuscript tables/figures and those in the Supporting Information, using labels such as "Table S1" for Supporting Information as appropriate.

C57. The Abstract mentions less than 5 grams of API required, however it needs a direct comparison with the “state-of-the-art” e.g. >75% savings in materials. This would highlight the impact of the technology in easily understandable terms.

Response: We have conducted a thorough comparison where feasible. As noted in our response to Comment C26, direct benchmarking across all stages remains challenging due to fundamental differences between our platform and state of the art approaches. Current approaches do not support digital optimisation of filler combinations without experimental blending and compaction work, nor do they offer integrated automation for powder handling, compaction, and testing. Given these limitations, we focused our comparison on the most directly comparable elements, for example, material usage for fixed excipient choice (binder/filler 1 and 2, lubricant and disintegrant) where meaningful equivalence could be established. Our aim was to provide a fair and informative assessment, while also emphasising the unique integration and automation capabilities of our platform that are not currently represented in state-of-the-art workflows.

We compared our work to two state-of-the-art approaches: Corrigan et al. and Joliffe et al. Both studies were conducted through industry-academia collaborations, reflecting the current state of the art across both sectors.

For comparison, we aligned two key factors: 1) consistent tablet weight across studies, and 2) similar overall objectives. It is important to note that these two studies focused exclusively on tablet porosity and tensile strength, without accounting for optimisation of blend flowability, an aspect addressed in our work.

This comparison demonstrates a 65% reduction in API usage with our platform compared to the approach reported by Corrigan et al.

In this comparative analysis, we report an API material usage of 2.65 g. The previously reported 5 g figure accounts for scenarios involving larger tablet sizes, high drug loadings, and minor material losses. For a fair comparison, we used exact values across all studies based on a defined tablet size and drug loading, while assuming similar powder loss across methods. The other studies did not report powder losses; however, such losses are also expected in manual operations, such as manual filling or shoe/paddle feeding.

Change: We have added a new section “RESULTS - Benchmarking and Performance Metrics, and added a new Section 6 to the Supporting Information to provide details about this comparison.

C58. Line 506: The reviewer found the use of versions confusing and unnecessary. Given the excessive length of the manuscript, the reviewer suggests removal of any reference to “version”, and focus on the latest and most up to date results.

Response: We have removed the use of versions in the main manuscript. This comparison has been moved to the Supporting Information.

Change: We have removed any reference to the different versions in the revised manuscript and only discuss the comparison of the version in Section 2 of the Supporting Information.

C59. The word “DataFactory” in the title is confusing. It maybe best to drop it; “Self-Driving Tableter” would be enough.

Response: We appreciate the reviewer’s suggestion regarding the title. However, we respectfully believe that the term “DataFactory” plays an important role in communicating the broader context and intent of our work. The concept of a DataFactory reflects a key architectural and functional aspect of the system – namely, its design as a cyber-physical platform that not only automates the process of tablet manufacturing but also continuously generates, aggregates, and utilises data in a feedback-driven loop to optimise performance and enable intelligent decision-making. This data-centric integration distinguishes the system from a conventional automated tablet press. The term “DataFactory” conveys this emphasis on digital infrastructure, and continuous improvement based on real-time analytics - aligning with modern paradigms of Quality by Digital Design.¹⁵ Therefore, we kept “DataFactory” in the title.

C60. Line 315 – what happened to the powder if they are neither diverted to waste nor reused? It is implied that it will be used by the process anyway. Clarify or remove.

Response: To clarify, there are only two possible outcomes for the powder. If the powder fails to meet the required quality criteria during testing, it is diverted to the waste container, from where it can be reused later after appropriate handling. If the powder meets the quality requirements, it proceeds directly to tableting. There are no scenarios where powder bypasses quality checks or is unaccounted for in the process.

C61. Line 316 – How are the tablets not meeting uniformity criteria identified?

Response: In the current implementation, the orchestration system monitors tablet weight uniformity in real-time as a proxy for content uniformity. Tablets falling outside of the predefined weight specification limits are automatically identified and discarded by the system.

However, the identification of tablets not meeting content uniformity criteria was not within the scope of the present research work. The authors acknowledge the importance of this aspect and are planning to apply NIR spectroscopy directly on tablets in future studies to assess and identify potential content uniformity issues in real-time.

C62. Line 358 – Is there a mode other than self-driving that the authors intended to compare?

Response: We have been able to run the Tableting DataFactory in a self-driving mode, where it is driven by Bayesian optimisation, and a manufacturing mode where it produces a certain number of tablets at a given condition.

Changes: *We have restructured the manuscript and revised Figure 1 to improve readability and avoid confusion regarding the different modes of operation.*

C63. Line 419 – Figures S8 and S9 have nothing to do with the context. Consequently, the comparison cannot be assessed by the reviewer.

Response: We included Figures S19 and S20 (previously Figures S8 and S9) because we believe they provide important supplementary information that may aid in further clarification or in-depth understanding of the system's performance. While they may not be central to the main narrative, we consider them useful for readers who may seek additional context or wish to explore comparative aspects beyond the core discussion.

References:

- 1 Wang, N., Sun, H., Dong, J. & Ouyang, D. PharmDE: A new expert system for drug-excipient compatibility evaluation. *International Journal of Pharmaceutics* **607**, 120962 (2021).
- 2 Wyttenbach, N., Birringer, C., Alsenz, J. & Kuentz, M. Drug-excipient compatibility testing using a high-throughput approach and statistical design. *Pharmaceutical development and technology* **10**, 499-505 (2005).
- 3 Vreeman, G. & Sun, C. C. A strategy to optimize precompression pressure for tablet manufacturing based on in-die elastic recovery. *International journal of pharmaceutics* **654**, 123981 (2024).
- 4 Markl, D. & Zeitler, J. A. A review of disintegration mechanisms and measurement techniques. *Pharmaceutical research* **34**, 890-917 (2017).
- 5 Leane, M., Pitt, K., Reynolds, G. & Group, M. C. S. W. A proposal for a drug product Manufacturing Classification System (MCS) for oral solid dosage forms. *Pharmaceutical development and technology* **20**, 12-21 (2015).
- 6 Leane, M. Manufacturing Classification System (MCS): Enabling Better Understanding of Oral Solid Dosage Formulation Design. *Pharmaceutical Development and Technology*, 1-3 (2024).
- 7 Sinka, I., Motazedian, F., Cocks, A. & Pitt, K. The effect of processing parameters on pharmaceutical tablet properties. *Powder Technology* **189**, 276-284 (2009).

- 8 Pitt, K. G., Webber, R. J., Hill, K. A., Dey, D. & Gamlen, M. J. Compression prediction accuracy from small scale compaction studies to production presses. *Powder Technology* **270**, 490-493 (2015).
- 9 Salehian, M. *et al.* A hybrid system of mixture models for the prediction of particle size and shape, density, and flowability of pharmaceutical powder blends. *International Journal of Pharmaceutics: X* **8**, 100298 (2024).
- 10 Yanes, D., Shinebaum, R., Papakostas, G., Reynolds, G. K. & Swainson, S. M. A pragmatic mixing model for the evaluation of powder flow properties of multicomponent pharmaceutical blends. *International Journal of Pharmaceutics: X*, 100339 (2025).
- 11 Owasit, A., Tripathi, S., Davé, R. & Young, J. Predicting Powder Blend Flowability from Individual Constituent Properties Using Machine Learning. *Pharmaceutical Research*, 1-19 (2025).
- 12 Wünsch, I., Finke, J. H., John, E., Juhnke, M. & Kwade, A. The influence of particle size on the application of compression and compaction models for tableting. *International journal of pharmaceutics* **599**, 120424 (2021).
- 13 Reynolds, G. K., Campbell, J. I. & Roberts, R. J. A compressibility based model for predicting the tensile strength of directly compressed pharmaceutical powder mixtures. *International Journal of Pharmaceutics* **531**, 215-224 (2017).
- 14 Corrigan, J. *et al.* An interaction-based mixing model for predicting porosity and tensile strength of directly compressed ternary blends of pharmaceutical powders. *International Journal of Pharmaceutics* **664**, 124587 (2024).
- 15 Mustoe, C. L. *et al.* Quality by digital design to accelerate sustainable medicines development. *International Journal of Pharmaceutics*, 125625 (2025).

Response to Reviewers for “Accelerated Medicines Development using a Digital Formulator and a Self-Driving Tableting DataFactory”

Faisal Abbas^{1^} & Mohammad Salehian^{1^}, Peter Hou¹, Jonathan Moores¹, Jonathan Goldie¹, Alexandros Tsioutsios¹, Theo Tait¹, Victor Portela², Quentin Boulay³, Roland Thiolliere³, Ashley Stark⁴, Jean-Jacques Schwartz⁴, Jerome Guerin⁴, Andrew G. P. Maloney⁵, Alexandru A. Moldovan⁵, Gavin K. Reynolds⁶, Jérôme Mantanus⁷, Catriona Clark¹, Paul Chapman², Alastair Florence¹, Daniel Markl^{1*}

[^] Authors contributed equally.

* Corresponding author; email: daniel.markl@strath.ac.uk

¹ CMAC, Strathclyde Institute of Pharmacy and Biomedical Science (SIPBS), University of Strathclyde, Glasgow, G1 1RD, UK

² Glasgow School of Art, Glasgow, G3 6RQ , UK

³ Medelpharm, ZAC des Malettes, 615 Rue du Chat Botté, 01700 Beynost, France

⁴ DEC Group, Chemin du Dévent 3, 1024 Ecublens, Switzerland

⁵ The Cambridge Crystallographic Data Centre, 12 Union Road, Cambridge, CB2 1EZ, UK

⁶ Sustainable Innovation & Transformational Excellence (xSITE), Pharmaceutical Technology & Development, Operations, AstraZeneca UK Limited, Macclesfield, SK10 2NA, UK

⁷ UCB S.A, 60 Allée de la Recherche, 1070 Brussels, Belgium

We would like to thank the reviewers for their interest in this work and for taking the time to thoroughly review the paper. All comments have been responded to, and the manuscript is revised accordingly.

Text Colour Key:

Reviewers Comment

Authors Response

Changes in Paper

Reviewer #1:

I have carefully reviewed the revised manuscript titled ‘Accelerated Medicines Development using a Digital Formulator and a Self-Driving Tableting DataFactory’. The authors have addressed most of my critical suggestions and have responded thoughtfully to the comments from other reviewers as well. While a few minor points remain, I am generally satisfied with the revisions and the authors’ clarifications.

I believe this paper represents a significant contribution to the field of medicines development, introducing innovative approaches that open up many avenues for future research and design. I therefore strongly recommend it for publication in Nature Communications.

Reviewer #2:

The authors well answered the reviewers' questions.

Reviewer #3:

The author team has properly addressed the comments from my previous review.

Review #5:

The authors’ extensive effort in revising the manuscript is recognized. The overall quality has been improved. The enhanced data and presentation elevated the manuscript as 1 or 2 publishable papers in a PharmSci or a process engineer journal. The reviewer recommends a rejection of the manuscript to Nature Communications. In addition to a continued disagreement with the authors on the NatComms worthy innovation, the manuscript falls short in scientific rigor and validated impact required. As the authors incorporated the clarity, visualization, and language issues suggested by the reviewer, the general concerns from the reviewer were either insufficiently addressed or deflected.

C1: The inclusion of new data and contents, while intended to address some of the previous issues, created new inconsistencies, a large number of new questions (e.g, the majority of the 43 figures in supplemental material were not clearly explained), and concerning patterns. The issues are too many to enumerate, with a few examples listed below.

Response: We have added further explanation to the Supplementary Information as requested. Most of the figures currently included were already present in the previous version, either in the main manuscript or in the Supplementary Information. As the reviewer did not specify which aspects require clarification (beyond the below comments), we can only speculate about what additional details might be helpful.

Changes: More explanation has been added to the sections 1, 3.1.4, 3.1.5, 4.3.2, 4.3.4, 4.2.3, and 6 in the supplementary information:

C2: As a major conclusion, 65% material reduction has been emphasized in the abstract, main innovation summary, and the first discussion point. The supporting data, however, is presented as the last section of the supplemental material, where the number of tablets manufactured from each method are substantially different. The reviewer disagrees with this conclusion, while concerns on the benchmarking objectivity at a principle level.

Response: We appreciate the reviewer’s careful assessment of the benchmarking analysis. We would like to clarify that the benchmarking section was placed at the end of the supplementary information to mirror the structure of the main manuscript, where the benchmarking discussion appears immediately before the general discussion section. For consistency between the manuscript and its supplementary information, we retained this relative positioning in the supplementary information.

Regarding the reviewer’s concern about the difference in the number of tablets manufactured using our approach (three tablets per pressure point with 4-5 pressure points) versus conventional methods¹⁻³ (typically 3-10 tablets with 8-10 pressure points), we agree that the absolute numbers differ. To provide a fair comparison, we benchmarked our approach against two state-of-the-art studies, including the method by Corrigan, et al.,⁴ which uses the smallest amount of material among currently published approaches. Corrigan, et al.⁴ performed 10 pre-determined pressure points with three tablets at each pressure point. Importantly, the difference in the number of pressure points and tablets is inherent to the innovation we are reporting rather than a limitation. The key contribution of our method is that, instead of producing and testing three tablets at several pre-determined pressure points, our Bayesian optimisation-guided process identifies pressure points adaptively leading to a reduced number of pressure points and tablets. This allows us to reliably model the tensile strength–porosity relationship using only 4-5 pressure points with three tablets per pressure point without compromising model accuracy. Therefore, the material reduction arises directly from the self-driving and data-efficient nature of the workflow, and the benchmarking reflects this contrast in experimental approach rather than a mismatch in study design.

Changes: We added the following information to Section 6 in the supplementary information to clarify this point:

“Corrigan, et al. performed 10 pre-determined pressure points with three tablets at each pressure point. Importantly, one of the key differences between our method and Corrigan, et al. is that, instead of producing and testing three tablets at several pre-determined pressure points, this study uses a Bayesian optimisation-guided process to identify pressure points adaptively leading to a reduced number of pressure points and tablets. This enables the calibration of a reliable model for the tensile strength–porosity relationship using only 4-5 pressure points with three tablets per pressure point without compromising model accuracy.”

C3: While Figure 3 was enhanced with more data, questions and concerns on the original data from the reviewer were ignored. Can the reviewer assume that the authors acknowledged that previous Figure 3 was wrong, and replaced it with the new Figure 3? New Figure 3b increased the range of tensile strength, though new Figure 3a decreased the range of porosity. What happened to the porosity data between 0.3-0.4? Were they thrown away because of a poorer fit? By the same token, should the reviewer question potential manipulation of other data?

Response: Thank you for raising this point. The increase in the tensile strength range is a consequence of expanding the validation dataset with additional formulations. These new, validated data points represented robust tablets with higher tensile strengths, which naturally correspond to lower porosity values. This expansion extended the upper bound of the tensile strength axis.

The decrease in porosity range is the direct physical outcome of implementing a systematic, automated data quality control pipeline after the ingestion of the original dataset to ensure that only physically plausible datapoints are included in the validation set. As we increased the validation dataset size (total of 546 data points), it became critical to automatically check with quality of the validation data. The main reasons for poor data quality are:

1. Physically implausible data points arising from experimental error, such as issues in blend preparation or incorrect compaction settings.
2. Undetected tablet defects, including lamination⁵, capping⁶, or cracking⁷, that affect measured properties. These defects can be difficult to identify visually. They typically manifest at very low or very high compression pressures / porosities, where porosity or tensile strength deviates from expected trends (i.e., porosity should steadily decrease with increasing compression force, and tensile strength should decrease with increasing porosity).

These data quality checks prevent the modelling framework from using data containing measurements of tablets with defects or data collection errors. This quality control pipeline was applied to the entire validation dataset and includes the following automated checks:

- Sanity check: Missing data (compression pressure, porosity, and tensile strength) identification and removal.
- Invalid values: Negative, zero, or Not-a-Number (NaN) compression pressure, porosity, and tensile strength values. Such values are physically implausible.
- Empirical Quality Check: Fitting compressibility (Kawakita⁸) and compactability (Ryshkewitch-Duckworth⁹) models to all tablet formulations and flagging the data with poor goodness-of-fit ($R^2 < 0.7$ for either model) as outliers to identify tablet defects. This conservative threshold ensures that only data exhibiting acceptable compression/compaction profile are included in model validation.

The pipeline identified two specific formulations from the initial validation dataset as outliers: one paracetamol (SP) tablet with anomalous compressibility behaviour (reflected in higher porosity values, which was mentioned by the reviewer) and one aspirin (AS) tablet with poor compactability fit (corresponding to lower porosity values) [see figures below]. Note that the number of validation data (546) reflects the dataset size after the automated quality control pipeline.

a) Formulation containing SP, MCC, MAN, CCS, MgSt b) Formulation containing AS (pure API tablet)

This quality control pipeline was applied automatically and systematically to all validation data using pre-defined, objective criteria. No data were selectively removed to improve model fit; rather, the pipeline ensures that physically implausible or experimentally problematic data do not compromise model evaluation.

Changes: We have enhanced the manuscript with an additional explanation and a schematic (Supplementary Information) that clearly demonstrates the automated data ingestion, quality control criteria, and the training/validation splitting process.

“To ensure the reliability and robustness of the process models, a systematic data quality control pipeline was implemented after ingesting the original dataset to ensure the physical validity of the training and validation data (Figure S7). The pipeline enforces data integrity through multiple automated stages:

- **Data Cleaning:** The dataset is screened for logging errors, removing entries with missing values, or non-physical values (negative, zero, or NaN) for compression pressure, porosity, and tensile strength.
- **Empirical Quality Check:** To filter out data resulting from measurements of tablets with defects or data collection errors such as over-compression⁷, capping⁶, or lamination⁵, the compressibility and compactability profiles of each formulation are fitted to the Kawakita and Ryshkewitch-Duckworth models, respectively. A user-defined goodness-of-fit threshold ($R^2 < 0.7$ in this study) is applied. Formulations failing to meet this conservative threshold are flagged as low-fit outliers and excluded from the original dataset to ensure the models are developed and validated using only physically valid data.

Figure S7: Schematic representation of the automated data ingestion, quality control, and training/validation framework.”

C4: Shortened abstract focused mainly on the problem statement with inflated significance that the reviewer finds trouble to agree, while failing to summarize key scientific innovation. This reflects the overarching lack of clear, to the point innovation from this paper. Piling up data and material does not necessarily make it more innovative, if not the opposite.

Response: We made a modification in the abstract to add more information about the innovation of our study. Given the scale of the work, a substantial amount of data and supporting material was generated and has been provided to ensure all relevant details are accessible to readers.

Changes: *We have modified the abstract.*

“In this work, we present an integrated platform for tablet formulation and process development that couples a Digital Formulator, an in-silico optimisation tool using a predictive material-to-tablet model and a curated excipient library, with a Self-driving Tableting DataFactory, which applies Bayesian optimisation within an automated, fully integrated per-tablet workflow spanning dosing, transport, compaction, and testing. The results of this platform demonstrate a reduction in the time from material characterisation to in-specification tablets to 6 hours and a reduction in API material use by 65% compared to current state-of-the-art methods.”

References:

- 1 Tait, T. *et al.* Empirical Model Variability: Developing a new global optimisation approach to populate compression and compaction mixture rules. *International Journal of Pharmaceutics* **662**, 124475 (2024).
- 2 Reynolds, G. K., Campbell, J. I. & Roberts, R. J. A compressibility based model for predicting the tensile strength of directly compressed pharmaceutical powder mixtures. *International Journal of Pharmaceutics* **531**, 215-224 (2017).
- 3 Vreeman, G. & Sun, C. C. Some properties and applications of the tableability equation. *International journal of pharmaceutics* **671**, 125246 (2025).
- 4 Corrigan, J. *et al.* An interaction-based mixing model for predicting porosity and tensile strength of directly compressed ternary blends of pharmaceutical powders. *International Journal of Pharmaceutics* **664**, 124587 (2024).
- 5 Mazel, V. & Tchoreloff, P. Lamination of pharmaceutical tablets: classification and influence of process parameters. *Journal of Pharmaceutical Sciences* **111**, 1480-1485 (2022).
- 6 Meynard, J., Amado-Becker, F., Tchoreloff, P. & Mazel, V. On the complexity of predicting tablet capping. *International Journal of Pharmaceutics* **623**, 121949 (2022).
- 7 Vreeman, G. & Sun, C. C. A strategy to optimize precompression pressure for tablet manufacturing based on in-die elastic recovery. *International journal of pharmaceutics* **654**, 123981 (2024).
- 8 Kawakita, K. & Tsutsumi, Y. An empirical equation of state for powder compression. *Japanese journal of applied physics* **4**, 56 (1965).
- 9 Duckworth, W. Discussion of ryshkewitch paper by winston duckworth. *J. Am. Ceram. Soc.* **36**, 68-69 (1953).